# Phc2 controls hematopoietic stem and progenitor cell mobilization from bone marrow by repressing *Vcam1* expression

Joonbeom Bae[1,8], Sang-Pil Choi[1,8], Kyoichi Isono[2], Ji Yoon Lee[3], Si-Won Park[1], Chang-Yong Choi[1], Jihye Han[1], Sang-Hoon Kim[1], Han-Hyoung Lee[1], Kyungmin Park[1], Hyun Yong Jin[4], Suk Jun Lee[5], Chung-Gyu Park [6], Haruhiko Koseki[7], Young Sik Lee [1] & Taehoon Chun [1]

The timely mobilization of hematopoietic stem and progenitor cells (HSPCs) is essential for maintaining hematopoietic and tissue leukocyte homeostasis. Understanding how HSPCs migrate between bone marrow (BM) and peripheral tissues is of great significance in the clinical setting, where therapeutic strategies for modulating their migration capacity determine the clinical outcome. Here, we identify an epigenetic regulator, Phc2, as a critical modulator of HSPC trafficking. The genetic ablation of *Phc2* in mice causes a severe defect in HSPC mobilization through the derepression of *Vcam1* in bone marrow stromal cells (BMSCs), ultimately leading to a systemic immunodeficiency. Moreover, the pharmacological inhibition of VCAM-1 in *Phc2*-deficient mice reverses the symptoms. We further determine that Phc2-dependent *Vcam1* repression in BMSCs is mediated by the epigenetic regulation of H3K27me3 and H2AK119ub. Together, our data demonstrate a cell-extrinsic role for Phc2 in controlling the mobilization of HSPCs by finely tuning their bone marrow niche.

[1] Department of Biotechnology, College of Life Sciences and Biotechnology, Korea University, Seoul 02841, Republic of Korea. [2] Laboratory Animal Center, Wakayama Medical University, Wakayama 641-8509, Japan. [3] Department of Biomedical Science, CHA University, Seongnam, Gyounggi-do 13488, Republic of Korea. [4] Department of Urology, School of Medicine, University of California, San Francisco, San Francisco, CA 94158, USA. [5] Department of Biomedical Laboratory Science, College of Health Science, Cheongju University, Cheongju-si 28503, Republic of Korea. [6] Department of Microbiology and Immunology, Cancer Research Institute and Xenotransplantation Research Center, Seoul National University College of Medicine, Seoul 03087, Republic of Korea. [7] Laboratory for Developmental Genetics, RIKEN Center for Integrative Medical Sciences, Yokohama 230-0045, Japan. [8] These authors contributed equally: Joonbeom Bae, Sang-Pil Choi. Correspondence and requests for materials should be addressed to T.C. (email: tchun@korea.ac.kr)

Hematopoietic stem and progenitor cells (HSPCs) are rare populations of cells that can sustain the hematopoietic system by continuously replenishing blood cells[1,2]. Postnatally, most HSPCs engraft and reside in the BM niche, but a small fraction of these cells is mobilized and migrate into the peripheral blood (PB) to reconstitute the hematopoietic system under specific signaling cues[1,3,4]. Therefore, identifying key regulators that critically control HSPC mobilization and understanding their underlying mechanisms have been of significant interest to facilitate BM transplantation efficiency and to therapeutically inhibit the malignant migration of the transformed hematopoietic cells into peripheral tissues[5]. Indeed, several BM niche proteins, including stem cell factor (kit ligand), CXCL12 (SDF-1α), and β1 integrin have been identified as key signaling components to control the maturation of HSPCs, their migration into the circulation and their homing to the BM by mediating the interaction between HSPCs and bone marrow stromal cells (BMSCs)[2,5–7]. However, the molecular pathways that regulate the expression of these BM niche proteins remain unclear.

Polycomb group (PcG) proteins function as transcriptional repressors of target genes by mainly modulating histone methylation[8]. These PcG proteins can be divided into two functionally distinct complexes: polycomb repressor complex 1 (PRC1) and polycomb repressor complex 2 (PRC2). PRC2 induces the formation of trimethylated histone H3 at lysine 27 (H3K27me3) through methyltransferase activity of Ezh proteins which are component proteins in PRC2 complex[9]. The canonical PRC1 then recognizes and binds to H3K27me3 to sustain the transcriptional repression of a target gene[8,10,11]. Both canonical and noncanonical PRC1 complexes can cause additional transcriptional silencing as E3 ubiquitin ligases by forming mono-ubiquitylated histone H2A at lysine 119 (H2AK119ub)[8,12,13].

In this study, we demonstrate that Phc2, a component of the canonical PRC1, regulates HSPC mobilization through the repression of *Vcam1* expression by enhancing both H3K27me3 and H2AK119ub in BMSCs. Therefore, Phc2 deficiency causes a severe HSPC mobilization defect via the derepression of *Vcam1* in BMSCs, and the pharmacological inhibition of VCAM-1 in BMSCs significantly reverses the symptoms of Phc2-deficient mice. These data demonstrate the critical cell-extrinsic role of Phc2 in controlling HSPC mobilization and provide the first evidence of epigenetic control over HSPC mobilization.

## Results

**Phc2 deficiency leads to a defect in HSPC mobilization.** As an initial step to elucidate the functional role of Phc2 during hematopoiesis, we characterized immune phenotypes of *Phc2−/−* (KO) mice[14] compared to those of *Phc2+/+* (wild-type, WT) mice. We observed macroscopic abnormalities in the thymus and peripheral lymphoid organs of KO mice compared to those of WT mice. Both the thymus and spleen of the KO mice maintained normal architecture, but these organs were smaller than those of the WT mice (Fig. 1a, b). Consistent with this observation, the absolute numbers of immune cells and their precursors in the PB, thymus, spleen, and liver of the KO mice were significantly reduced compared to those of the WT mice (Tables 1 and 2, Fig. 1c). Notably, the absolute number of early T-cell precursors (ETPs) in the thymus or immature B cells in the spleen was significantly lower in the KO mice relative to the WT mice (Table 1).

Unexpectedly, the absolute numbers and relative ratios of HSPCs, early progenitor cells, and immune cells in the BM were similar among all examined mice (Table 1 and Fig. 1c, d). Additionally, the loss of *Phc2* did not influence the cell cycle progression or apoptosis of HSPCs (Fig. 1e, f), and both the WT and KO BM-resident HSPCs exhibited no difference in terms of their ability to generate multipotential or myeloid progenitor cells (Fig. 1g). The absolute numbers of WT and KO B cells in the BM at each developmental stage were not significantly different either (Fig. 1h). Given that no functional defect was evident in the HSPCs of the BM but that there was a shortage of ETPs and immature B cells from the BM, we postulated that a Phc2 deficiency could lead to a systemic immunodeficiency due to a defect in HSPC migration into the circulation.

To test this hypothesis, we examined the numbers of circulating HSPCs from WT and KO mice using a colony-forming unit (CFU) assay. The absolute numbers of clonogenic progenitors in the PB and spleen were decreased by approximately half in the KO mice compared to the WT mice, whereas the absolute numbers of clonogenic progenitors in the BM were comparable between the WT and KO mice (Supplementary Fig. 1). To confirm this result, we performed in vivo HSPC mobilization assays using two commonly used mobilization regimens for therapy, granulocyte colony-stimulating factor (G-CSF) and AMD3100 (CXCR4 antagonist)[15–18]. Five days after G-CSF treatment, the absolute numbers of white blood cells (WBCs) and splenocytes from the KO mice remained significantly lower than those from the WT mice (Fig. 2a, b). The spleen size of the KO mice also remained much smaller than that of the WT mice after G-CSF treatment (Fig. 2b). Consistent with these results, both the frequencies and absolute numbers of Lin−Sca-1+c-kit+ cells (LSK cells) in the PB and spleen were significantly lower in the KO mice than in the WT mice (Supplementary Fig. 2). Likewise, the absolute numbers of clonogenic progenitors in the PB and spleen of the KO mice were significantly reduced compared to those of the WT mice (Fig. 2c). However, the frequencies and absolute numbers of LSK cells in the BM were comparable between the WT and KO mice (Supplementary Fig. 2). Moreover, the absolute numbers of clonogenic progenitors in the BM were not significantly different between the WT and KO mice (Fig. 2c). When we used an AMD3100 as a mobilizing agent of LSK, we still observed that the migration of KO LSK into the periphery is statistically decreased compared with that of WT LSK (Fig. 2d, e). However, the defective migration of KO LSK after treatment of AMD3100 is relatively modest when we compared with results from G-CSF treatment (Fig. 2d, e).

To remove circulating HSPCs and visualize neo-HSPC mobilization from BM, we further treated WT and KO mice with 5-fluorouracil (5-FU), a chemotherapeutic agent that depletes circulating HSPCs and promotes HSPC release from BM during hematopoietic repopulation[19,20]. The absolute numbers of WBCs and splenocytes recovered from KO mice were lower than those recovered from WT mice starting on day 8 after 5-FU treatment (Fig. 2f, g). The most prominent differences in the absolute numbers of WBCs and splenocytes or the size of the spleen between WT and KO mice was observed on day 16 after 5-FU treatment (Fig. 2f, g). On day 16 after 5-FU treatment, the frequency and the absolute number of recovered LSK cells and the absolute number of recovered clonogenic progenitors in the PB and spleen were significantly lower in KO mice than in WT mice, whereas those in the BM were comparable between the WT and KO mice (Fig. 2h, Supplementary Fig. 3). Taken together, these results indicate that Phc2 deficiency causes a defect in HSPC mobilization from the BM into the periphery.

**_Phc2−/−_ BMSCs cause a defect in HSPC mobilization.** To identify which cell type in the BM niche contributed to the defect in HSPC release into the circulation of KO mice, we adoptively transferred LSK cells isolated from WT CD45.1 mice into

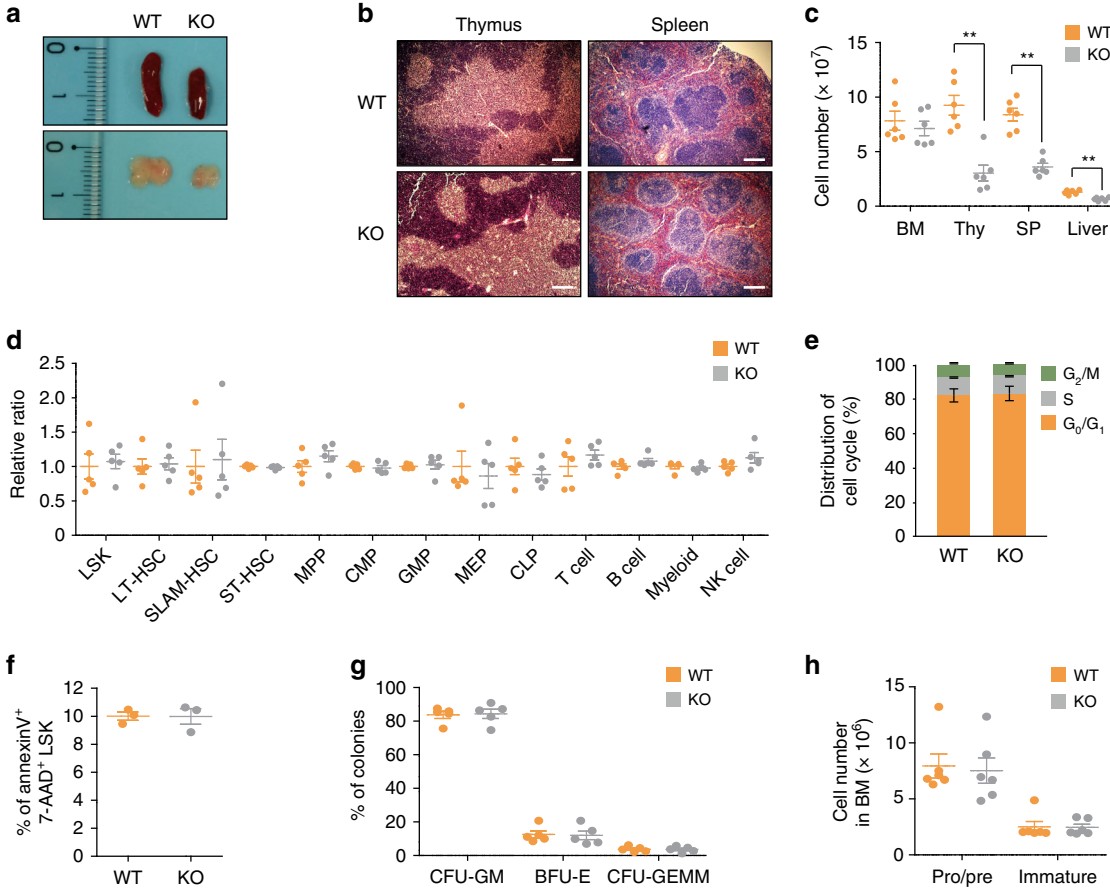

**Fig. 1** Phc2 deficiency leads to immune cell deficiency without intrinsic defects in HSPCs. **a** Representative images of spleens and thymi from WT and KO mice. $n = 6$. **b** Normal architecture of the thymus and spleen of the KO mouse. Representative hematoxylin and eosin staining images of spleens and thymi from WT and KO mice. $n = 6$. Scale bars: 200 μm. **c** Absolute number of immune cells in the BM, thymus (Thy), spleen (SP), and liver of WT or KO mice. $n = 6$. **d** Relative ratio of BM-resident hematopoietic lineage cells between WT and KO mice. $n = 5$. LSK, Lin−Sca-1+c-kit+ cells; LT-HSC, long-term HSC; SLAM-HSC, Lin−CD41−CD48−CD150+ cells; ST-HSC, short-term HSC; MPP, multipotent progenitors; CMP, common myeloid progenitors; GMP granulocyte/monocyte progenitors; MEP, megakaryocyte/erythrocyte progenitors; CLP, common lymphoid progenitors; Myeloid, myeloid cells. **e** Cell cycle status of BM-resident LSK cells from WT and KO BM. $n = 4$. **f** Percentage of apoptosis for BM-resident LSK cells from WT and KO mice. $n = 3$. **g** BM-resident clonogenic progenitors from WT and KO mice were assessed by CFU assays. $n = 5$. CFU-GM, CFU-granulocyte and/or macrophage; BFU-E, erythroid burst-forming units, CFU-GEMM, CFU-granulocyte/erythrocyte/monocyte/megakaryocyte. **h** Absolute number of BM-resident B cell subpopulations from WT and KO mice ($n = 6$). Pro/pre, pro-B cells/pre-B cells; Immature, immature B cells. Statistical significance was assessed by two-tailed Student's $t$-test. **$P < 0.01$. All data are presented as the mean ± SEM. Source data are provided as a Source Data File

irradiated WT or KO CD45.2 recipient mice (Fig. 3a). KO recipients showed impaired repopulation of donor-derived immune cells in the thymus and spleen (Fig. 3b, c). However, the absolute numbers of LSK cells and lineage-committed progenitors (Lin−c-kit+ cells; LK cells) in the BM of KO recipients were comparable to those in the WT recipients (Fig. 3b–d). To confirm these observations, irradiated WT CD45.1 mice were reconstituted with LSK cells isolated from WT or KO CD45.2 mice (Fig. 3e). The origin of the donor LSK cells did not affect the repopulation of leukocytes in WT recipients (Fig. 3f, g). The origin of the donor LSK cells did not affect the absolute number of LSK cells or LK cells in the BM of WT recipients either (Fig. 3h). We also performed a serial competitive BM repopulation assay to further determine whether KO HSPCs had an intrinsic defect. We found that KO HSPCs had the same ability to reconstitute immune cells as WT HSPCs (Supplementary Fig. 4). This strongly suggests that the defect in HSPC release into the circulation in KO mice is caused by an extrinsic factor from BMSCs within BM niches.

To confirm this result, we performed an in vivo homing assay with CFSE-labeled WT and KO LSK cells. The number of CFSE+ LSK cells homing to the BM was higher among the KO recipients than among the WT recipients (Fig. 4a). Such an increase in the short-term migration of CFSE+ LSK cells into the BM of KO recipients led to fewer CFSE+ LSK cells homing to the PB and spleen of the KO recipients (Fig. 4a). Accordingly, the frequency of clonogenic progenitors in the BM of the KO recipients was increased, whereas the frequencies of clonogenic progenitors in the PB and spleen of the KO recipients were decreased compared to those of the WT recipients (Fig. 4b, Supplementary Fig. 5). We next measured the capacity of WT and KO LSK cells to transmigrate across their BMSCs using a trans-stromal migration assay. The number of LSK cells transmigrating across KO BMSCs was smaller than the number of LSK cells transmigrating across WT BMSCs (Fig. 4c, Supplementary Fig. 6). In contrast, the number of LSK cells adhering to KO BMSCs was greater than the number of LSK cells adhering to WT BMSCs (Fig. 4d, Supplementary Fig. 6). Collectively, these data demonstrate that an extrinsic factor from BMSCs causes HSPCs to adhere more strongly to BMSCs, ultimately leading to the defective mobilization of HSPCs from the BM into the periphery in KO mice.

**Table 1 Absolute numbers of immune cells and their precursors in _Phc2_$^{+/+}$ and _Phc2_$^{-/-}$ mice[a]**

| Tissue | Population | Surface phenotype | _Phc2_$^{+/+}$ | _Phc2_$^{-/-}$ | p-value | n |
|---|---|---|---|---|---|---|
| BM (×10$^4$) | LSK | Lin$^-$Sca-1$^+$c-kit$^+$ | 3.68 ± 0.75 | 3.62 ± 0.52 | 0.953 | 5 |
| | LT-HSC | Lin$^-$Sca-1$^+$c-kit$^+$Flt3$^-$CD34$^-$ | 0.56 ± 0.12 | 0.58 ± 0.04 | 0.813 | 5 |
| | SLAM-HSC | Lin$^-$Sca-1$^+$c-kit$^+$CD48$^-$CD150$^+$ | 0.65 ± 0.25 | 0.73 ± 0.26 | 0.869 | 5 |
| | ST-HSC | Lin$^-$Sca-1$^+$c-kit$^+$Flt3$^-$CD34$^+$ | 1.35 ± 0.18 | 1.50 ± 0.22 | 0.615 | 5 |
| | MPP | Lin$^-$Sca-1$^+$c-kit$^+$Flt3$^+$CD34$^+$ | 1.53 ± 0.18 | 1.89 ± 0.42 | 0.449 | 5 |
| | CMP | Lin$^-$Sca-1$^-$c-kit$^+$CD16/32$^-$CD34$^+$ | 10.57 ± 2.24 | 10.01 ± 2.23 | 0.862 | 5 |
| | GMP | Lin$^-$Sca-1$^-$c-kit$^+$CD16/32$^+$CD34$^+$ | 14.25 ± 1.35 | 14.75 ± 2.59 | 0.87 | 5 |
| | MEP | Lin$^-$Sca-1$^-$c-kit$^+$CD16/32$^-$CD34$^-$ | 7.72 ± 1.67 | 6.83 ± 1.05 | 0.663 | 5 |
| | CLP | Lin$^-$Sca-1$^-$c-kit$^+$CD127$^+$ | 0.90 ± 0.12 | 1.01 ± 0.12 | 0.566 | 5 |
| | T cell | CD3ε$^+$ | 183.57 ± 40.24 | 178.34 ± 16.64 | 0.907 | 5 |
| | B cell | B220$^+$ | 1317.83 ± 157.03 | 1357.77 ± 112.11 | 0.837 | 5 |
| | Pro/pre B cell | B220$^+$IgM$^-$IgD$^-$ | 793.78 ± 106.97 | 752.46 ± 112.88 | 0.796 | 6 |
| | Immature B cell | B220$^+$IgM$^+$IgD$^-$ | 250.34 ± 47.52 | 245.04 ± 28.29 | 0.926 | 6 |
| | Myeloid cell | CD11b$^+$ | 3480.12 ± 466.76 | 3260.69 ± 368.49 | 0.722 | 5 |
| | NK cell | NK1.1$^+$ | 83.95 ± 11.97 | 93.97 ± 16.35 | 0.634 | 5 |
| Thymus (×10$^6$) | ETP | Lin$^-$CD25$^-$CD44$^+$c-kit$^+$ | 0.11 ± 0.01 | 0.03 ± 0.00 | <0.001 | 5 |
| | DN1 | CD4$^-$CD8$^-$CD44$^+$CD25$^-$ | 0.35 ± 0.08 | 0.08 ± 0.01 | 0.011 | 5 |
| | DN2 | CD4$^-$CD8$^-$CD44$^+$CD25$^+$ | 0.15 ± 0.04 | 0.04 ± 0.00 | 0.036 | 5 |
| | DN3 | CD4$^-$CD8$^-$CD44$^-$CD25$^+$ | 1.55 ± 0.40 | 0.44 ± 0.09 | 0.027 | 5 |
| | DN4 | CD4$^-$CD8$^-$CD44$^-$CD25$^-$ | 2.35 ± 0.71 | 0.43 ± 0.09 | 0.027 | 5 |
| | DP | CD4$^+$CD8$^+$ | 66.42 ± 4.50 | 17.69 ± 2.49 | <0.001 | 5 |
| | CD4 SP | CD4$^+$CD8$^-$ | 10.45 ± 1.28 | 2.72 ± 0.51 | 0.001 | 5 |
| | CD8 SP | CD4$^-$CD8$^+$ | 4.94 ± 0.46 | 1.41 ± 0.38 | <0.001 | 5 |
| Spleen (×10$^6$) | T cell | CD3ε$^+$ | 20.72 ± 1.91 | 7.72 ± 0.73 | <0.001 | 5 |
| | CD4 + T cell | CD4$^+$ | 12.18 ± 0.99 | 4.59 ± 0.38 | <0.001 | 5 |
| | CD8 + T cell | CD8$^+$ | 8.12 ± 0.15 | 3.21 ± 0.34 | <0.001 | 5 |
| | B cell | B220$^+$ | 40.24 ± 2.81 | 15.89 ± 1.13 | <0.001 | 5 |
| | Immature B cell | B220$^+$IgM$^+$IgD$^-$ | 8.52 ± 0.66 | 3.85 ± 0.31 | <0.001 | 6 |
| | Mature B cell | B220$^+$IgM$^+$IgD$^+$ | 24.93 ± 0.75 | 11.67 ± 1.12 | <0.001 | 6 |
| | Macrophage | CD11b$^+$F4/80$^+$ | 4.89 ± 1.03 | 1.75 ± 0.35 | 0.021 | 5 |
| | Dendritic cell | CD11b$^+$CD11c$^+$ | 2.55 ± 0.67 | 0.83 ± 0.17 | 0.038 | 5 |
| | NK cell | NK1.1$^+$ | 0.14 ± 0.02 | 0.05 ± 0.01 | 0.003 | 5 |

_LSK_ Lin$^-$Sca-1$^+$c-kit$^+$ cell, _LT-HSC_ long term-hematopoietic stem cell, _SLAM-HSC_ Lin$^-$CD41$^-$CD48$^-$CD150$^+$cell, _ST-HSC_ short term-hematopoietic stem cell, _MPP_ multipotent progenitor, _CMP_ common myeloid progenitor, _GMP_ granulocyte-monocyte progenitor, _MEP_ megakaryocyte-erythroid progenitor, _CLP_ common lymphoid progenitor; _ETP_ early T-cell precursor, _DN_ double negative cell, _SP_ single positive cell
Source data are provided as a Source Data File
[a]p-values were generated by two-tailed unpaired Student's _t_-test. All data represent the mean ± SEM

**Table 2 Complete blood counts in _Phc2_$^{+/+}$ and _Phc2_$^{-/-}$ mice[a]**

| | _Phc2_$^{+/+}$ | _Phc2_$^{-/-}$ | p-value |
|---|---|---|---|
| WBC (×10$^3$/μL) | 8.47 ± 0.56 | 5.71 ± 0.53 | 0.005 |
| RBC (×10$^6$/μL) | 10.53 ± 0.24 | 10.25 ± 0.18 | 0.361 |
| Hb (g/dL) | 16 ± 0.37 | 15.67 ± 0.21 | 0.448 |
| HCT (%) | 47.17 ± 0.95 | 45.33 ± 0.49 | 0.117 |
| MCV (fL) | 44.72 ± 0.38 | 44.23 ± 0.30 | 0.342 |
| MCH (pg) | 15.20 ± 0.18 | 15.18 ± 0.27 | 0.960 |
| MCHC (g/dL) | 33.98 ± 0.15 | 34.31 ± 0.39 | 0.449 |
| PLT (×10$^3$/μL) | 876.67 ± 125.26 | 1071.67 ± 30.27 | 0.161 |

_WBC_ white blood cells, _RBC_ red blood cells, _Hb_ hemoglobin, _HCT_ hematocrit, _MCV_ mean corpuscular volume, _MCH_ mean corpuscular hemoglobin, _MCHC_ MCH concentration, _PLT_ platelets
Source data are provided as a Source Data File
[a]p-values were generated by two-tailed unpaired Student's _t_-test. n = 6. All data represent the mean ± SEM

**PcG proteins repress _Vcam1_ gene expression in BMSCs.** To understand the molecular mechanisms through which PcG strengthens the adhesion properties of HSPCs to BMSCs, we first investigated the compositions and architectures of BM niches in WT and KO mice and measured the expression levels of candidate genes that are known to be involved in these processes. Endothelial cells, mesenchymal stem cells, and osteoblasts are major BMSCs found in BM niches[21–23]. Importantly, neither a change in BMSC composition nor an abnormal architecture was found in the KO BM niches (Supplementary Figs. 7 and 8). Next, we examined the expression patterns of soluble factors and cell surface molecules that might regulate HSPC mobilization in WT and KO BMSCs[2,7] (Fig. 5a, Supplementary Fig. 9). Intriguingly, both the mRNA and protein levels of VCAM-1 were significantly increased in KO BMSCs compared to WT BMSCs (Fig. 5a, b). Flow cytometry analysis further confirmed that the expression of VCAM-1 was significantly higher in KO BMSCs than in WT BMSCs (Fig. 5c). Consistent with this observation, _Vcam1_ mRNA levels were significantly elevated in activated KO peritoneal macrophages compared to those of WT (Supplementary Fig. 10).

Taken together, these results indicate that Phc2 may repress _Vcam1_ expression in BMSCs. To further examine Phc2-dependent _Vcam1_ repression, we knocked down Phc2 expression in BMSCs (OP9 cells) and concurrently monitored _Vcam1_ expression levels under treatment with TNF-α, a well-known inducer of _Vcam1_ transcription[24] (Fig. 5d). As expected, TNF-α treatment induced the expression of _Vcam1_ at 3 h post-treatment. However, this was significantly exacerbated when Phc2 was depleted, and thus, the VCAM-1 level was not dampened up to 12 h post-TNF-α treatment (Fig. 5e, f). Overall, these results suggest that Phc2 represses the expression of _Vcam1_ in BMSCs at the transcriptional level.

To demonstrate a role for Phc2 in the transcriptional repression of _Vcam1_, we performed a chromatin immunoprecipitation (ChIP) assay with seven primer pairs covering the

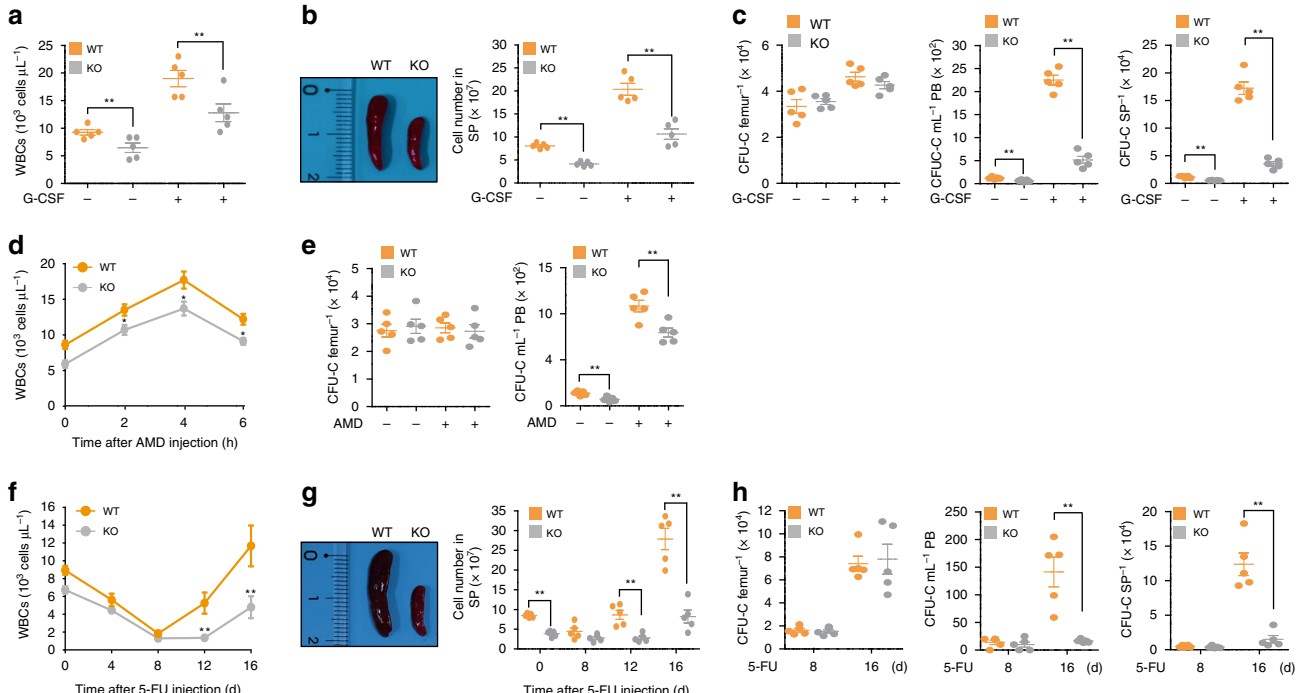

**Fig. 2** Impaired mobilization of KO HSPCs from the BM into the periphery. **a** Comparison of WBC counts between WT and KO mice on day 5 after G-CSF treatment. $n = 5$. **b** Representative image of spleens from WT and KO mice (left) and absolute numbers of splenocytes (right) for WT and KO mice on day 5 after G-CSF treatment. $n = 5$. **c** CFU-C in the BM (femur), PB, and spleen (SP) of WT and KO mice on day 5 after G-CSF treatment. $n = 5$. **d** Comparison of WBC counts between WT and KO mice 3 h after AMD3100 treatment. $n = 5$. **e** CFU-C in the BM (femur) and PB of WT and KO mice 3 h after AMD3100 treatment. $n = 5$. **f** Comparison of WBC counts between WT and KO mice at indicated time points after 5-FU treatment. $n = 5$. **g** Representative image of the spleen from a WT or KO mouse (left) and absolute number of splenocytes (right) for WT or KO mice at indicated time points after 5-FU treatment. $n = 5$. **h** CFU-C in the BM (femur), PB, and spleen (SP) of WT and KO mice at indicated time points after 5-FU treatment. $n = 5$. "−", before treatment with G-CSF or AMD3100; " + ", after treatment with G-CSF or AMD3100. Statistical significance was assessed by two-tailed Student's $t$-test. **$P < 0.01$. All data are presented as the mean ± SEM. Source data are provided as a Source Data File

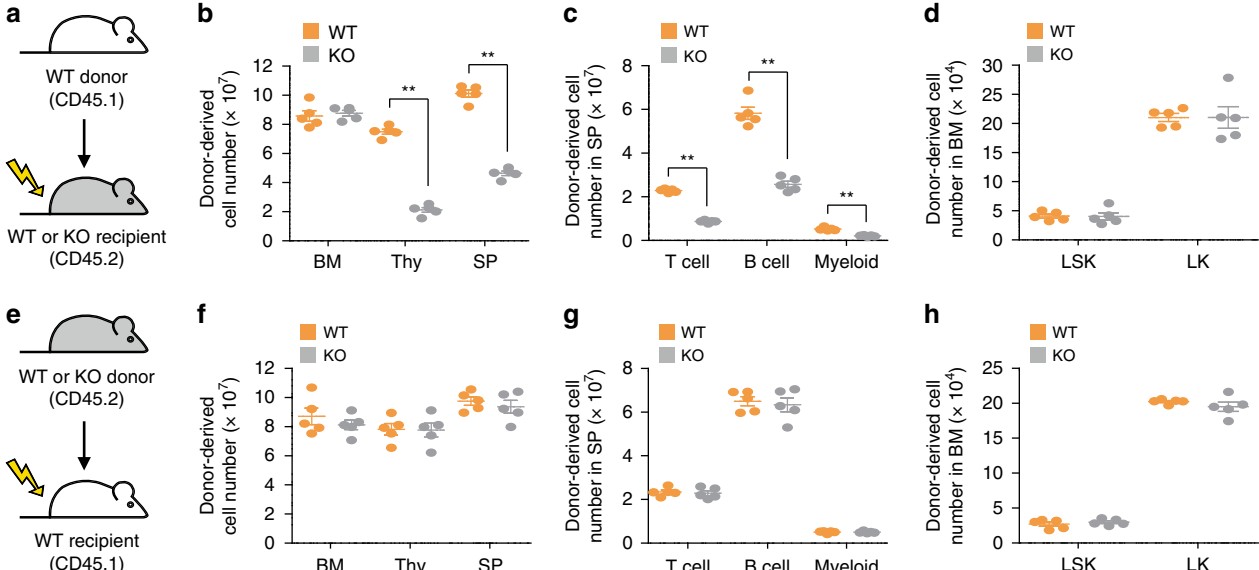

**Fig. 3** An extrinsic factor from BMSCs is responsible for the impaired mobilization of KO HSPCs from the BM into the periphery. **a–d** Lethally irradiated WT and KO CD45.2 mice were transplanted with LSK cells from WT CD45.1 mice. $n = 5$ per group. **a** Schematic representation of LSK cell adoptive transfer. **b** Absolute number of donor-derived cells in the BM, thymus (Thy), and spleen (SP) of recipients. **c** Absolute number of donor-derived T cells, B cells, and myeloid cells (Myeloid) in the spleen of recipients. **d** Absolute number of donor-derived LSK cells (LSK) and LK cells (LK) in the BM of recipients. **e–h** Lethally irradiated WT CD45.1 mice were transplanted with LSK cells from WT or KO CD45.2 mice. $n = 5$ per group. **e** Schematic representation of LSK cell adoptive transfer. **f** Absolute numbers of donor-derived cells in the BM, Thy, and SP of recipients. **g** Absolute numbers of donor-derived T cells, B cells, and myeloid cells in the spleen of recipients. **h** Absolute numbers of donor-derived LSK and LK cells in the BM of recipients. Statistical significance was assessed by two-tailed Student's $t$-test. **$P < 0.01$. All data are presented as the mean ± SEM. Source data are provided as a Source Data File

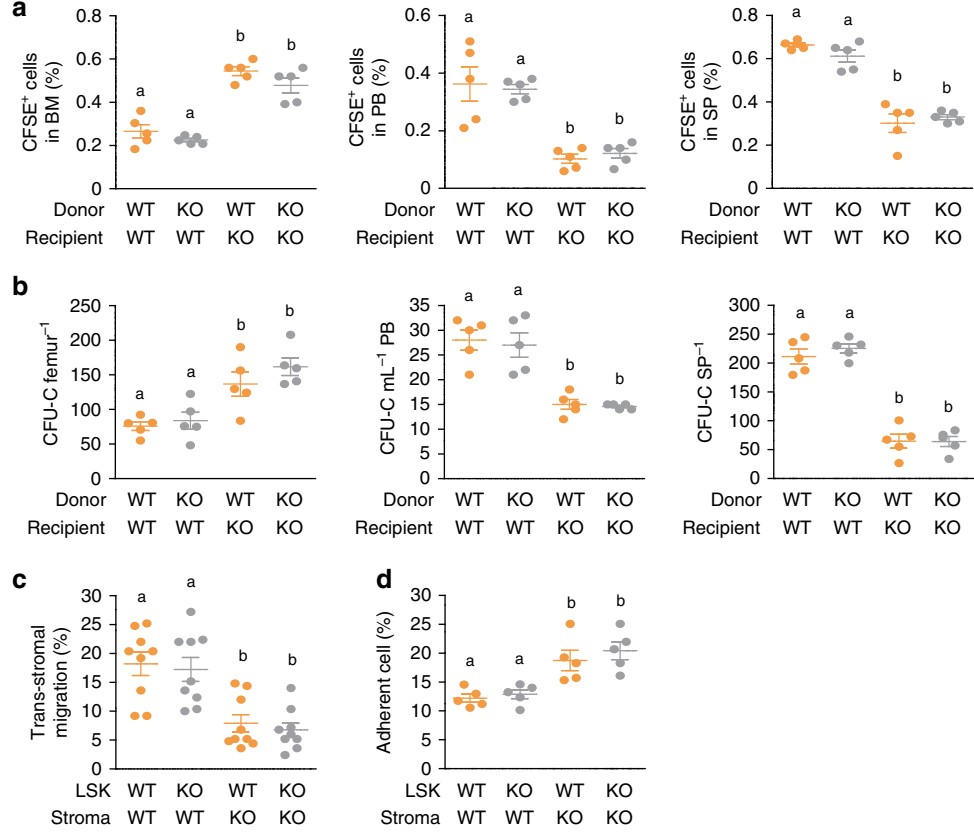

**Fig. 4** Phc2 deficiency leads to impaired HSPC mobilization from the BM into the periphery via stronger adhesion between HSPCs and BMSCs in BM niches. **a**, **b** CFSE-labeled LSK cells from WT and KO mice were intravenously injected into lethally irradiated recipient mice. $n = 5$ per group. **a** The frequencies of CFSE$^+$ donor cells were measured in the BM, PB and spleen (SP) of recipients at 16 h after injection. **b** The frequencies of donor-derived clonogenic progenitors (CFU-C) homing to the BM (femur), PB and spleen (SP) of recipients at 16 h after injection. **c** Relative ratio of migrated WT or KO LSK cells (LSK) through WT or KO BMSCs (Stroma). $n = 9$ per group. **d** Relative ratio of adhered WT or KO LSK to WT or KO stroma. $n = 5$ per group. Statistical significance was assessed by one-way ANOVA with Tukey HSD analysis. Mean values not sharing the same superscript letter ($^a$, $^b$) differ significantly at $P < 0.05$. All data are presented as the mean ± SEM. Source data are provided as a Source Data File

putative Phc2 binding sites located in the *Vcam1* gene (Fig. 6a). This analysis revealed the strong binding of Phc2 to the promoter region of the *Vcam1* locus (Fig. 6b, primers #2 and #3). Interestingly, direct binding by other members of the canonical PRC1 family, such as Bmi1, Ring1b, and Cbx7, was also found around the same region covered by primers #2 and #3 (Fig. 6c). Consistent with this, Bmi1, Ring1b, and Cbx7 binding to these regions were significantly decreased in the absence of Phc2 (Fig. 6c). The acetylation of histone H3 at lysine 9/14 (H3K9ac), trimethylation of histone H3 at lysine 4 (H3K4me3), H3K27me3, and H2AK119ub were observed around the same region covered by primers #2 and #3 (Fig. 6c). H3K9ac and H3K4me3 were increased, whereas H3K27me3 and H2AK119ub were decreased in KO BM cells compared to WT BM cells (Fig. 6c). Furthermore, the treatment of BMSCs (OP9 cells) with GSK126, an Ezh2 inhibitor[25], led to the upregulation of *Vcam1* expression in a dose-dependent manner (Fig. 6d, e and Supplementary Fig. 11). Consistent with these observations, the binding of Phc2, Bmi1, and Ring1b to the regions covered by primers #2 and #3 was decreased in GSK126-treated BMSCs (OP9 cells) (Fig. 6f).

Next, we then asked whether GSK126 treatment could mimic Phc2 deficiency in vivo. Different dose of GSK126 was administered intraperitoneally into WT mice. Sixteen hours after GSK injection, BMSCs were isolated and analyzed for VCAM-1 expression. We also measured the capacity of WT LSK cells to transmigrate across their GSK126-treated BMSCs using a trans-stromal migration assay. Results clearly showed that GSK126 treatment increased both VCAM-1 expression in BMSCs and adherence of LSKs onto BMSCs in a dose-dependent manner (Supplementary Fig. 12). Therefore, the canonical PRC1 containing Phc2 can repress *Vcam1* expression through histone modifications, such as by enhancing H3K27me3 and H2AK119ub, in BMSCs.

**Anti-VCAM-1 Ab treatment restores *Phc2*$^{-/-}$ HSPC mobilization.** Next, we examined whether blocking the interaction between VCAM-1 and VLA-4 could restore HSPC mobilization in KO mice. We first attempted to measure the mobilization capacity of WT LSK cells across KO BMSCs after treatment with neutralizing antibodies (Abs) against VCAM-1 and/or VLA-4. Treatment with either Ab restored the trans-stromal migration activity of LSK cells across KO BMSCs to a level similar to that of LSK cells across WT BMSCs (Fig. 7a). Additionally, there was no significant difference in the relative ratio of LSK cell adhesion to KO or WT BMSCs after treatment with the neutralizing Abs (Fig. 7b). The in vivo homing assay further confirmed that blocking the interaction between VCAM-1 and VLA-4 could restore LSK cell mobilization from the KO BM into the PB and spleen (Fig. 7c, d and Supplementary Fig. 13). We then tested whether treatment of the neutralizing VCAM-1 Ab could rescue steady-state HSPC mobilization in KO mice. Results clearly demonstrated that the administration of the neutralizing VCAM-

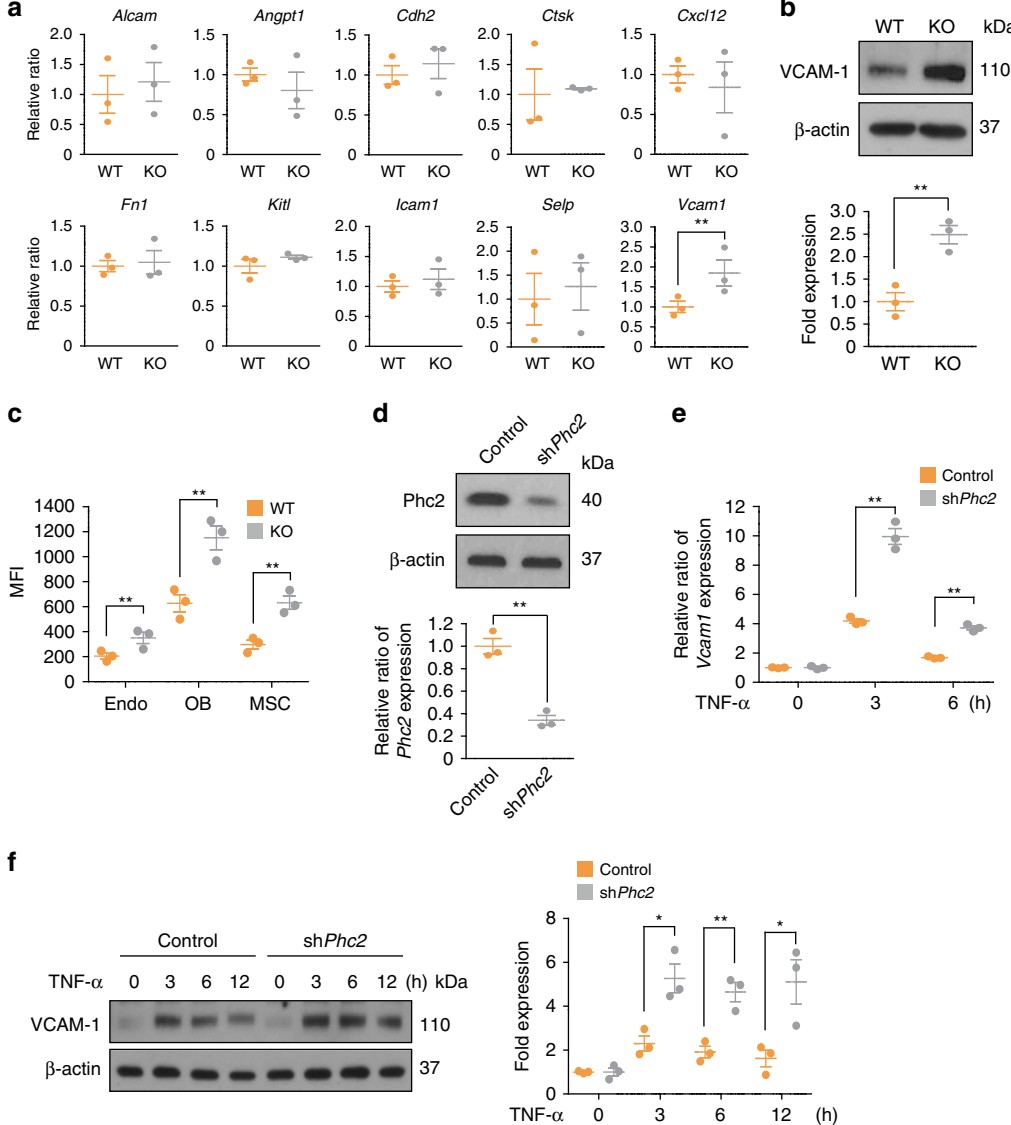

**Fig. 5** Phc2 inhibits VCAM-1 expression in BMSCs. **a** mRNA expression patterns of genes related to HSPC mobilization in BM niches from WT and KO mice were accessed by qRT-PCR. $n = 3$. **b** Lysates from WT and KO BMSCs were immunoblotted to analyze VCAM-1 expression. $n = 3$. **c** Mean fluorescence intensity (MFI) of VCAM-1 expression in BMSCs from WT and KO mice analyzed by flow cytometry. $n = 3$ Endo, endothelial cells; OB, osteoblasts; MSC, mesenchymal stem cells. **d** Control plasmid (Control) and sh*Phc2* plasmid (sh*Phc2*) were transfected into BMSCs (OP9 cells). qRT-PCR (bottom) and immunoblot (top) analyses were then performed to measure the Phc2 expression levels. $n = 3$. **e**, **f** Control transfectants (Control) and sh*Phc2* transfectants (sh*Phc2*) were treated with TNF-α for the indicated time period. qRT-PCR (**e**) and immunoblot (**f**) analyses were then performed to measure VCAM-1 expression. Relative fold change of each *Vcam1* mRNA expression from Control or sh*Phc2* was calculated relative to the basal level of that in unstimulated Control or sh*Phc2*, respectively. $n = 3$. Statistical significance was assessed by two-tailed Student's t-test. *$P < 0.05$; **$P < 0.01$. All data are presented as the mean ± SEM. Source data are provided as a Source Data File

1 Ab in KO mice successfully reconstitutes HSPC mobilization from BM into periphery (Fig. 7e–g).

We also performed a G-CSF-, AMD3100-, or 5-FU-induced LSK cell mobilization assay using WT and KO mice treated with the neutralizing VCAM-1 Ab. There were no significant differences in the absolute numbers of WBCs and splenocytes between the WT and KO mice treated with anti-VCAM-1 on day 5 after the G-CSF treatment and on day 16 following the 5-FU challenge (Fig. 8a, b, f, g). Similarly, the absolute number of LSK cells and the clonogenic progenitor frequency in the BM, PB, and spleen were comparable between WT and KO recipients treated with the neutralizing Ab on day 5 after the G-CSF treatment and on day 16 following the 5-FU challenge (Fig. 8c, h, Supplementary Figs. 14 and 15). Consistent with these results, the defective

HSPC migration into the PB of the KO mice was reversed by treating the mice with anti-VCAM-1, as observed in the HSPC mobilization assay using AMD3100 alone or in combination with G-CSF (Fig. 8d, e and Supplementary Fig. 16).

To further reinforce these observations, we assessed the significance of VCAM-1 blockage when the migration defect of HSPCs are driven by extrinsic microenvironment which mimics the phenotype associated with Phc2 KO mice. First, we adoptively transferred LSK cells isolated from WT CD45.1 mice into irradiated WT or KO CD45.2 recipient mice and then performed the same G-CSF-induced LSK cell mobilization assay using recipient mice treated with the neutralizing VCAM-1 Ab (Supplementary Fig. 17). Consistent to the HSPC mobilization assay in KO mice, WT cells reconstituted in irradiated KO

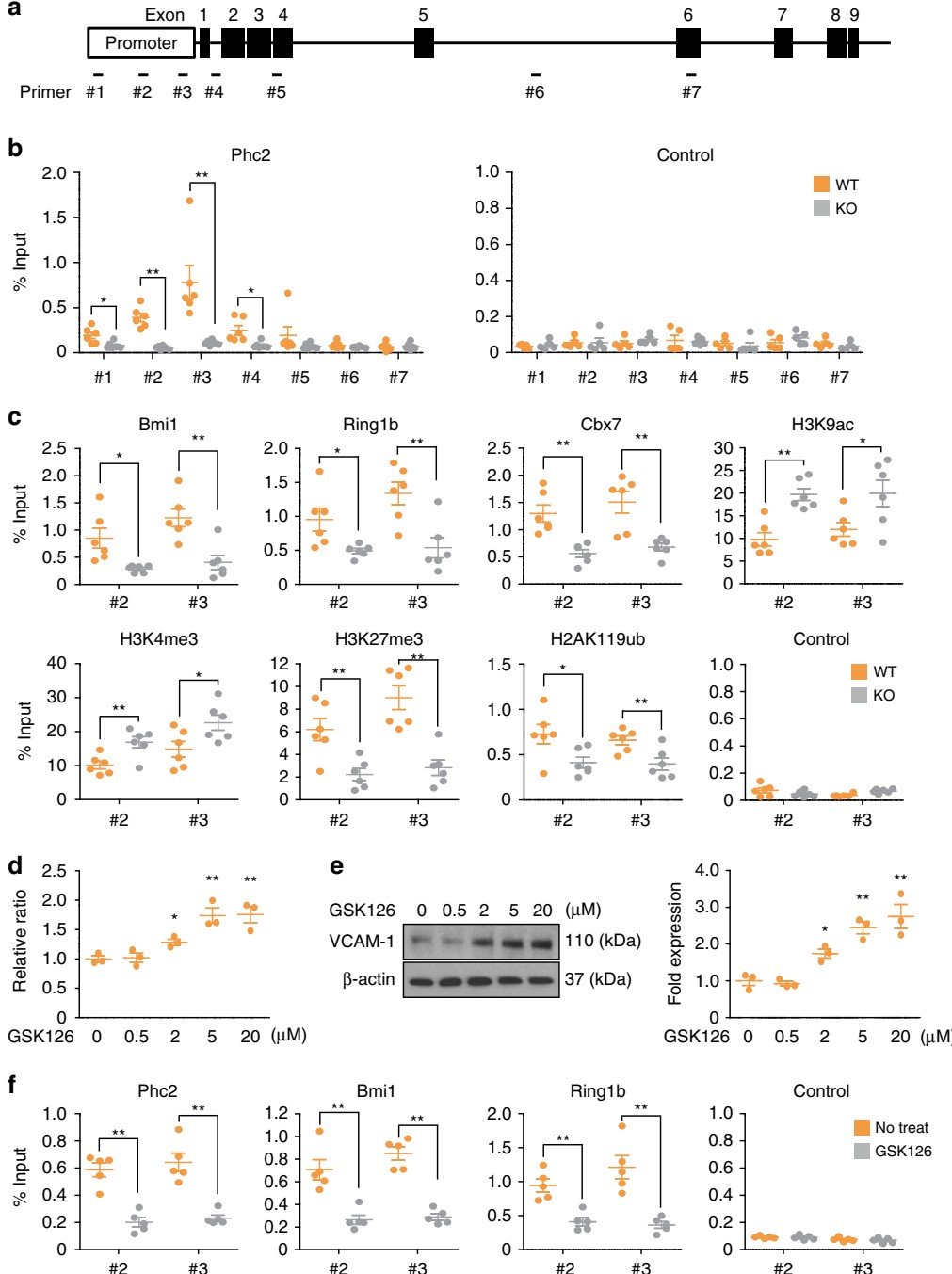

**Fig. 6** Phc2 binds to the *Vcam1* gene locus and suppresses *Vcam1* gene expression through histone modifications. **a** Schematic representation of the *Vcam1* gene locus. The locations of primers #1 to #7 and exons are indicated. **b** ChIP was performed with WT and KO BM cell lysates using anti-Phc2 Ab (Phc2) or a control Ab (Control). DNA recovered from ChIP was then analyzed by qPCR. $n = 5$. **c** ChIP was performed using the same samples with indicated Abs. DNA recovered from ChIP was then analyzed by qPCR. $n = 6$. Control, ChIP using each isotype control Ab. **d**, **e** BMSCs (OP9 cells) were treated with various concentrations of GSK126 for 48 h. qRT-PCR (**d**) and immunoblot (**e**) analyses were then performed to measure the VCAM-1 expression levels. $n = 3$. **f** BMSCs (OP9 cells) were treated with GSK126 (5 μM) for 48 h. ChIP was performed using cell lysates with indicated Abs. DNA recovered from ChIP was then analyzed by qPCR. $n = 5$. Control, ChIP using each isotype control Ab. Statistical significance was assessed by two-tailed Student's $t$-test. $*P < 0.05$; $**P < 0.01$. All data are presented as the mean ± SEM. Source data are provided as a Source Data File

recipient cells were only partially rescued the mobilization defects when the single G-CSF agent was treated (Supplementary Fig. 17). However, the administration of the neutralizing VCAM-1 Ab in combination with G-CSF in KO recipient mice completely reconstituted WT donor HSPC mobilization from BM into periphery, the same degree to WT recipient mice (Supplementary Fig. 17). Taken together, these results demonstrate that blocking

the interaction between VCAM-1 and VLA-4 can restore HSPC migration from the BM into the periphery of KO mice.

## Discussion

In this study, we demonstrated that Phc2 binds to the *Vcam1* locus to repress its transcription by enhancing H3K27me3 and

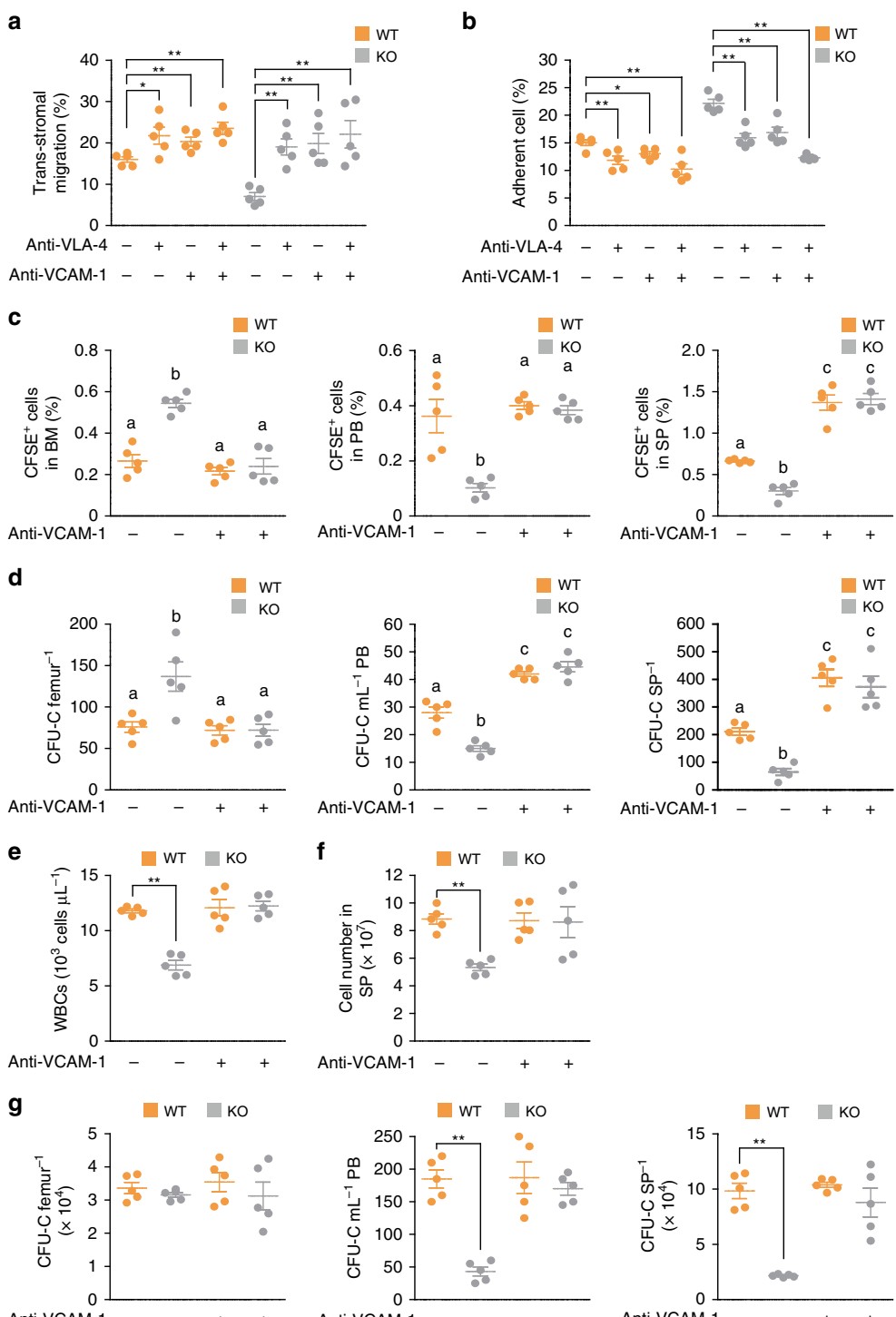

H2AK119ub in BMSCs. In the absence of Phc2, the increased *Vcam1* expression in BMSCs strengthened the interaction between HSPCs and BMSCs, compromising timely HSPC mobilization from the BM into the periphery, which in turn led to a systemic immunodeficiency. Consistent with these observations, treatment with a neutralizing Ab against VCAM-1 in Phc2-deficient mice restored HSPC mobilization from the BM into the periphery. These data firmly establish a causative role for the epigenetic regulation of VCAM-1 in regulating HSPC mobilization, as a key downstream mediator of Phc2 functions.

Recent genetic evidence from conditional VCAM-1- or VLA-4-deficient mice further supports our model of the VLA-4/VCAM-1

pathway as downstream of Phc2 function, and as expected from our results with Phc2-deficient mice overexpressing VCAM-1, VCAM-1- or VLA-4 deficient mice exhibit increased circulating progenitors and immature B cells, and impaired HSPC homing to BM[26–30]. The BM homing defect of HSPCs in these mice became obvious when the mice were treated with 5-FU or G-CSF[26,27]. However, the Phc2-deficient mice still displayed the migration defect of HSPCs from the BM into the periphery after treatment with these mobilization agents.

The identification of a single key mediator of Phc2 function may have important clinical implications. Similar to previous observations with conditional VCAM-1- or VLA-4-deficient

**Fig. 7** Blocking the interaction between VCAM-1 and VLA-4 restores steady-state HSPC mobilization in KO mice. **a** Relative ratio of migrated WT LSK cells (LSK) through WT or KO BMSCs in the presence of anti-VLA-4 Ab and/or anti-VCAM-1 Ab. $n = 5$ per group. Statistical significance was assessed by two-tailed Student's $t$-test. *$P < 0.05$; **$P < 0.01$. **b** Relative ratio of adhered WT or KO LSK to KO BMSCs in the presence of anti-VLA-4 Ab and/or anti-VCAM-1 Ab. $n = 5$ per group. Statistical significance was assessed by two-tailed Student's $t$-test. *$P < 0.05$; **$P < 0.01$. **c**, **d** CFSE-labeled LSK cells from WT mice were intravenously injected into lethally irradiated WT or KO mice. Anti-VCAM-1 Ab or its isotype control Ab (2 mg kg$^{-1}$) was intravenously injected into WT and KO recipient mice 1 h before adoptive cell transfer. $n = 5$ per group. **c** The frequencies of CFSE$^+$ donor cells were measured in the BM, PB, and, spleen (SP) of recipients at 16 h after adoptive cell transfer. **d** The frequencies of donor-derived clonogenic progenitors (CFU-C) homing to the BM (femur), PB and SP of recipients at 16 h after adoptive cell transfer. Statistical significance was assessed by one-way ANOVA with Tukey HSD analysis. Mean values not sharing the same superscript letter (a, b, c) differ significantly at $P < 0.05$. **e–g** Anti-VCAM-1 Ab or its isotype control Ab (2 mg kg$^{-1}$) was intravenously injected into WT and KO mice for 3 days. $n = 5$ per group. **e** Comparison of WBC counts between WT and KO mice on day 3 after Ab treatment. **f** Absolute number of splenocytes for WT and KO mice on day 3 after Ab treatment. **g** CFU-C in the BM (femur), PB, and SP of WT and KO mice on day 3 after Ab treatment. Statistical significance was assessed by two-tailed Student's $t$-test. **$P < 0.01$. All data are presented as the mean ± SEM. WT, WT recipient mice; KO, KO recipient mice. "−", treatment with an isotype control Ab; "+", treatment with anti-VCAM-1 Ab. Source data are provided as a Source Data File

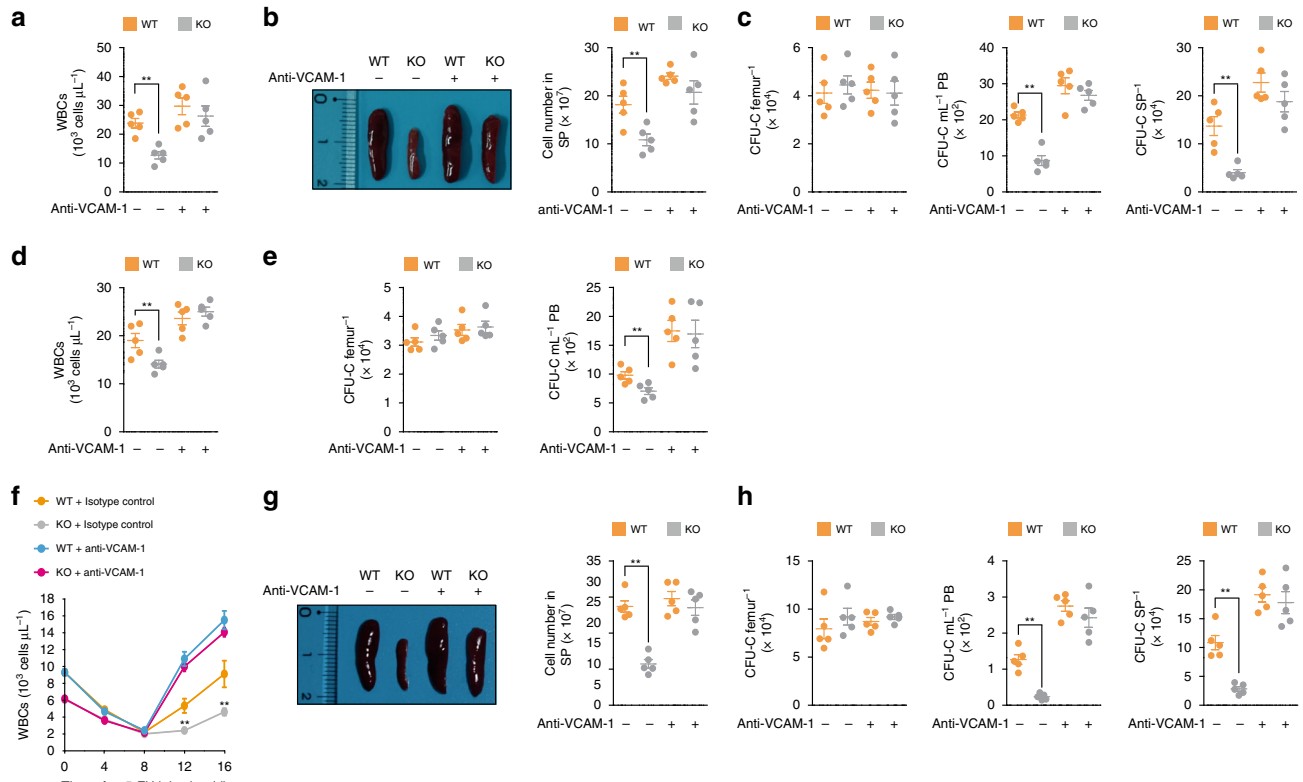

**Fig. 8** Neutralizing interaction between VCAM-1 and VLA-4 reverses the defect involving regimen-induced HSPC mobilization in KO mice. **a–c** G-CSF-induced HSPC mobilization assays with anti-VCAM-1 Ab. $n = 5$ per group. **a** WBC counts from experimental animals until day 5 after G-CSF and Ab treatment. **b** Representative image of spleens (left) and absolute number of splenocytes from experimental animals on day 5 after G-CSF and Ab treatment (right). **c** CFU-C in the BM (femur), PB, and spleen (SP) from experimental animals on day 5 after G-CSF and Ab treatment. **d**, **e** WBC counts (**d**) and CFU-C (**e**) in PB from experimental animals 3 h after AMD3100 and Ab treatment. $n = 5$ per group. **f–h** 5-FU-induced HSPC cell mobilization assays with anti-VCAM-1 Ab. $n = 5$ per group. **f** Comparison of WBC counts between WT and KO mice at indicated time points after 5-FU and Ab treatment. **g** Representative image of spleen from a WT and KO mouse (left) and absolute number of splenocytes (right) of WT and KO mice on day 16 after 5-FU and Ab treatment. **h** CFU-C in the BM (femur), PB, and spleen (SP) of WT and KO mice on day 16 after 5-FU and Ab treatment. "−", treatment with an isotype control Ab; "+", treatment with anti-VCAM-1 Ab. Statistical significance was assessed by two-tailed Student's $t$-test. **$P < 0.01$. All data are presented as the mean ± SEM. Source data are provided as a Source Data File

mice[26–30], we found an abnormality of only steady-state HSPC mobilization in Phc2-deficient mice. Additionally, treatment with the anti-VCAM-1 Ab or anti-VLA-4 Ab in primates or mice does not cause any other physiological change[31,32]. Combined evidence may suggest a dominant role for VCAM-1 in steady-state HSPC migration due to the VCAM-1 expression pattern. In steady-state conditions, the BMSC is the only cell type that constitutively expresses VCAM-1 at physiological levels, although vascular endothelial cells and other immune cell populations

express minimally detectable levels of VCAM-1[33–38]. Furthermore, with a proof-of-concept experiment, we successfully demonstrated that blocking VCAM-1 in BM niches critically reverses the HSPC mobilization defect in Phc2-deficient mice. Therefore, along with other recent data[39–43], our results suggest that modulating the interaction between VCAM-1 and VLA-4 in BM niches provides potential strategies to develop HSPC mobilizing agents or anti-leukemia drugs. Therefore, it will be interesting to observe whether a specific inhibitor that blocks Phc2

function is useful for keeping malignant cells from hematopoietic lineages within the BM.

To our knowledge, we are the first to provide evidence that a systemic immunodeficiency can be caused by a defect in a particular epigenetic regulator that represses the expression of a certain cell adhesion molecule in BMSCs. Our study also demonstrates the molecular mechanism through which the epigenetic regulation of extrinsic factors can help finely tune the interaction between HSPCs and BMSCs within BM niches during HSPC mobilization. Although several genes that are involved in hematopoiesis have been identified as targets for epigenetic regulation, the epigenetic effect on HSPC mobilization has not been elucidated. Notably, the vast majority of previously identified genes play important roles in defining the intrinsic property of HSCs and the development of leukemia[8,44,45].

Like deficiencies in other epigenetic regulators, a single PcG gene deficiency in mice mostly causes early embryonic lethality or a limited life span with severe developmental defects, including an intrinsic defect in HSCs[8,46,47]. This characteristic hinders the search for additional roles of polycomb proteins in the regulation of immune phenomena aside from their known functions in HSC differentiation. Similar to other members of the PcG protein family, Phc2 is expressed in various tissues and cells and regulates the transcription of $Cdkn2a$ ($p16^{INK4a}$ and $p19^{ARF}$) and $hox$ genes through direct associations with chromatin[8,14]. However, Phc2-deficient mice are fertile and exhibit less severe defects associated with the abnormal axial skeleton and Peyer's patch[14,48]. Moreover, the systemic immunodeficiency observed in Phc2-deficient mice is not caused by an intrinsic defect in HSC function, but instead, the Phc2 deficiency causes a defect in HSPC mobilization by enhancing the expression of VCAM-1 in BMSCs. This may suggest that most of the Phc2 functions in mammals are dispensable or can be replaced by other components of the canonical PRC1 due to a structural heterogeneity of PRC1 or functional redundancy of PcG proteins. In fact, each of the mammalian PRC2 or PRC1 subunits has several paralogs[8–13]. For example, Phc2 has two paralogs, Phc1 and Phc3[8–13]. Previous observation revealed that Phc1 deficiency causes a defect in HSC's self-renewal activity[49,50]. Therefore, Phc1, but not phc2, might be critical for HSC maintenance and function. Consistent with this idea, the expression and epigenetic pattern of $Cdkn2a$ ($p16^{INK4a}$ and $p19^{ARF}$) in KO BM cells showed no significant differences compared to those in WT BM cells (Supplementary Fig. 18). Another example of nonredundant function by each paralog within PRC2 or PRC1 components might be found in Ezh proteins or Cbx proteins, respectively[51–55]. Among Ezh proteins, Ezh1 is critical for self-renewal activity of HSCs, whereas Ezh2 is dispensable for proper HSC function[51,52]. Also, Cbx7 is the only paralog which is necessary for self-renewal activity of HSC among Cbx proteins[53,54].

In our molecular model, PRC2 containing Ezh2 with methyltransferase activity initiates H3K27me3 on the $Vcam1$ locus in BMSCs because the Ezh2 inhibitor (GSK126) treatment of BMSCs increases the $Vcam1$ mRNA levels by negating the recognition of H3K27me3 by the canonical PRC1 complex containing Phc2 on the same locus. The canonical PRC1 then recognizes and binds to H3K27me3 to maintain $Vcam1$ gene suppression. The recognition of H3K27me3 on the $Vcam1$ locus by the canonical PRC1 is dependent on $Phc2$ expression. Consistently, we observed reductions in the binding of other canonical PRC1 components to the $Vcam1$ locus when Phc2 was not expressed. Our study also revealed that the H2AK119ub activity of the canonical PRC1 on the $Vcam1$ locus was dependent on $Phc2$ expression, which is consistent with a recent observation that the interaction between Bmi1 and Phc2 is critical for the H2AK119ub activity of the canonical PRC1[55].

In conclusion, the results of our study demonstrated Phc2-regulated HSPC mobilization through the direct repression of $Vcam1$ gene expression in BMSCs. These findings may offer a novel approach to manipulating the epigenetic patterns of genes involved in HSPC mobilization, leading to new therapeutic strategies for BM transplantation and leukemia treatment.

## Methods

**Experimental mice and phenotypic analysis**. $Phc2^{+/-}$ ($Phc2$ heterozygote) mice with a C57BL/6 background were used in this study[14]. C57BL/6 and B6 CD45.1 mice were obtained from The Jackson Laboratory. WT and KO mice were bred from $Phc2^{+/-}$ mice. All animals received proper care in accordance with the National Institutes of Health Guide for the Care and Use of Laboratory Animals. The study protocol was approved by the Institutional Animal Care and Use Committee of Korea University (protocol numbers: KUIACUC-20110104-3, KUIACUC-20141002-4, and KUIACUC-20160927-3).

**Cell culture**. BMSCs were obtained and maintained from WT and KO mice[56]. Briefly, bone marrow (BM) cells were isolated from the femurs and tibias of 8-week-old mice and cultured in Dulbecco's modified Eagle medium (Welgene LM 001-05) supplemented with 10% fetal bovine serum (FBS; Welgene S 001–01), 10% horse serum (Welgene S 104–01), 2 mM L-glutamine (Welgene LS 002–01), 2 mM sodium pyruvate (Welgene LS 002–02), and 100 U ml$^{-1}$ penicillin/streptomycin (Welgene LS 202–02) at 37 °C and 5% $CO_2$. Non-adherent cells were removed by washing with PBS after overnight incubation. Adherent cells were cultured and fresh medium was added every 2 days. After 4 weeks of culture, cells were harvested by incubation with trypsin/EDTA (Welgene LS 015–01) for 2 min at 37 °C and plated onto culture dishes. At two weeks after the first passage, cells were harvested and cultured under the same conditions for subsequent passages (passages 2–7). OP9 BMSC line (ATCC CRL-2749) was maintained in alpha MEM (Welgene LM 008-02) supplemented with 20% FBS and 1% penicillin/streptomycin at 37 °C and 5% $CO_2$. For induction of VCAM-1, OP9 cells were treated with TNF-α (100 ng ml$^{-1}$; R&D Systems 410-MT) for the indicated time period. To inhibit Ezh activity, OP9 cells were treated with various concentrations of GSK126, an Ezh2 inhibitor (Xcessbio Biosciences M60071-2), for 48 h.

**PB count, flow cytometry analysis, and cell sorting**. PB samples from each mouse were collected by retro-orbital bleeding and analyzed using a Coulter LH 780 Hematology Analyzer (Beckman Coulter). For flow cytometry, single cell suspensions were prepared from the BM, PB, thymus, and spleen of each mouse. After removing red blood cells, cells were stained with MACS buffer at 4 °C in the presence of Fc Block Ab (BD Biosciences 553141, dilution 1:100). Abs purchased from BD Biosciences, Biolegend or Thermo Fisher Scientific were used to detect the following cell surface markers by flow cytometry: B220 (BD Biosciences 553090, dilution 1:100), CD3ε (Thermo Fisher Scientific 11-0031-63, dilution 1:100), CD4 (BD Biosciences 553729, dilution 1:100), CD8 (BD Biosciences 553030, dilution 1:100), CD11b (Thermo Fisher Scientific 11-0112-41, dilution 1:100), CD11c (BD Biosciences 553802, dilution 1:100), CD16:32 (Thermo Fisher Scientific 14-0161-81, dilution 1:100), CD25 (Thermo Fisher Scientific 17-0251-81, dilution 1:100), CD31 (BD Biosciences 561814, dilution 1:100), CD34 (BD Biosciences 560238, dilution 1:100), CD44 (BD Biosciences 561861, dilution 1:100), CD45 (BD Biosciences 550539, dilution 1:100), CD48 (Biolegend 103412, dilution 1:100), CD51 (BD Biosciences 551187, dilution 1:100), CD106 (BD Biosciences 553332, dilution 1:100), CD127 (Thermo Fisher Scientific 45-1271-80, dilution 1:100), CD150 (Biolegend 115922, dilution 1:100), c-kit (Thermo Fisher Scientific 17-1171-81, dilution 1:100), F4/80 (Thermo Fisher Scientific MF48004, dilution 1:100), Flt3 (Thermo Fisher Scientific 46-1351-80, dilution 1:100), IgD (BD Biosciences 562022, dilution 1:100), IgM (BD Biosciences 550676, dilution 1:100), NK1.1 (BD Biosciences 557391, dilution 1:100), and Sca-1 (BD Biosciences 558162, dilution 1:100). After washing several times with PBS, stained cells were resuspended in PBS and analyzed by flow cytometry using a FACSCalibur with CellQuest software (BD Biosciences). Analysis and sorting of HSPCs (Lin$^-$Sca-1$^+$c-kit$^+$ cells; LSK cells) and early progenitor cells were performed using MACS cell separation system and FACSAria Fusion Cell Sorter (BD Biosciences)[57]. Depletion of specific lineage cells was achieved with a lineage-specific Ab cocktail and anti-biotin microbeads (Miltenyi Biotec 130-090-858, dilution 1:5). Gating strategy for LSK cell sorting[58] was depicted in Supplementary Fig. 19.

**Histological analysis and immunoblotting**. For histological analysis[59], tissue samples were fixed with 10% formalin (Sigma, HT5011) and embedded in paraffin. Then, sections measuring 5 μm were cut using a Leica CM1800 cryostat (Leica Microsystems), and air dried at room temperature. After dry, slides containing each tissue sample were stained with hematoxylin (Merck 1.05174.0500) and eosin (Merck 109844) per the manufacturer's protocol.

For immunoblot analysis[59], cells were lysed with RIPA lysis buffer containing 50 mM Tris, pH 7.4, 150 mM NaCl, 1% NP-40 (Sigma, 74385), 0.5% sodium deoxycholate (Sigma, D6750), 0.1% SDS (Sigma, L4509) with 200 μg ml$^{-1}$ of phenylmethylsulfonyl fluoride (Sigma, P7626), phosphatase inhibitor cocktail

(Sigma, P0044) and protease inhibitor cocktail (Millipore, 535140). The cell lysates were then resolved by 12% SDS-polyacrylamide gel electrophoresis, transferred onto Immobilon P membranes (Millipore, IPVH00010), and immunoblotted with anti-VCAM-1 Ab (Santa Cruz Biotechnology, Inc. sc-8304, dilution 1:1000), anti-H3K27me3 Ab (EMD Millipore 07-449, dilution 1:1000), or anti-histone H3 Ab (Cell Signaling Technology 9715, dilution 1:1000) coupled with goat anti-rabbit IgG-HRP (Santa Cruz Biotechnology, Inc. sc-2004, dilution 1:2500), and anti-Phc2 Ab (mouse IgG, dilution 1:200)[14] or anti-β-actin Ab (Sigma-Aldrich A5441, dilution 1:25,000) coupled with goat anti-mouse IgG-HRP (Santa Cruz Biotechnology, Inc. sc-2005, dilution 1:2500). Immunoreactive bands were visualized using an ECL solution (Thermo Fisher Scientific 34080). To quantify the relative protein expression, immunoreactive bands were normalized to levels of β-actin. The relative band intensity of protein expression was quantified using ImageJ software (National Institutes of Health). Uncropped blots can be found in the Source Data.

**Cell cycle analysis and measurement of apoptosis.** BM-resident LSK cells from 8-week-old mice were isolated and incubated in 70% ethanol overnight at 4 °C. After washing with PBS, cells were stained with propidium iodide (Sigma-Aldrich P4170) for cell cycle analysis. To measure apoptosis, LSK cells were stained with 7-AAD (BD Biosciences 559925, dilution 1:20) and Annexin V-FITC (BD Biosciences 556547, dilution 1:20). After washing with PBS, stained cells were analyzed by flow cytometry.

**CFU assay.** Mononuclear cells isolated from the BM, PB, and spleen were plated onto Methocult GF M3434 (StemCell Technologies 03444) in 35 mm cell culture dishes. Cells were then incubated at 37 °C and 5% $CO_2$. After 7 days of incubation, type and number of colonies were determined[60].

**In vivo HSPC mobilization assays.** For G-CSF-induced mobilization assays[15], recombinant human G-CSF (Amgen Filgrastim Neupogen) was diluted in PBS with 0.1% BSA (GenDEPOT A0100-050). G-CSF or vehicle was administered by daily subcutaneous injection to 8-week-old mice at a dose of 250 μg kg$^{-1}$ day$^{-1}$ for 5 days. For AMD3100-induced mobilization assays[17], AMD3100 (Sigma-Aldrich A5602) was diluted in PBS and administered subcutaneously to 8-week-old mice at 5 mg kg$^{-1}$. For G-CSF and AMD3100-induced mobilization assays[61], G-CSF or vehicle was administered by daily subcutaneous injection to 8-week-old mice at a dose of 250 μg kg$^{-1}$ day$^{-1}$ for 5 days. One h before final G-CSF dose, AMD3100 administered subcutaneously to the same group of mice at 5 mg kg$^{-1}$. Three h after the final G-CSF or AMD3100 dose, mice were sacrificed to assess the absolute numbers of nucleated cells in the PB and spleen. The absolute numbers of LSK cells in the BM, PB, and spleen were determined by flow cytometry and CFU assays. To block the VLA-4 and VCAM-1 interaction, anti-VCAM-1 (BD Biosciences 553330) Ab or the respective isotype control (2 mg kg$^{-1}$ day$^{-1}$; BD Biosciences 553926) was administered intravenously every day.

For 5-FU-induced mobilization assays[62], 5-FU (Sigma-Aldrich F6627) was administered intravenously to 8-week-old mice at 200 mg kg$^{-1}$. The absolute numbers of nucleated cells in the PB and spleen were counted on days 4, 8, 12, and 16 after 5-FU treatment. On day 16 after 5-FU administration, mice were sacrificed, and the absolute numbers of Lin$^-$Sca-1$^+$c-kit$^+$ cells (LSK cells) in the BM, PB, and spleen were determined by flow cytometry and CFU assays. To block the VLA-4 and VCAM-1 interaction, mice were injected with anti-VCAM-1 Ab or the respective isotype control (2 mg kg$^{-1}$ day$^{-1}$) from day 8 to day 15 after 5-FU administration.

For anti-VCAM-1 Ab-induced mobilization assays[63], anti-VCAM-1 Ab or the respective isotype control administered by daily subcutaneous injection to 8-week-old mice at a dose of 2 mg kg$^{-1}$ day$^{-1}$ for 3 days. Eight hours after the final Ab dose, mice were sacrificed, and the absolute numbers of nucleated cells in the PB and spleen were counted. The absolute numbers of LSK cells in the BM, PB, and spleen were determined by CFU assays described above.

**LSK transplantation.** LSK cells (1 × 10$^5$) isolated from 8-week-old donor mice were injected intravenously into lethally irradiated (10 Gy) 8-week-old recipient mice. Twelve weeks after the LSK transfer, recipient mice were sacrificed and analyzed for reconstitution of donor immune cells. Donor cells and recipient cells were discriminated by flow cytometry using anti-CD45.2 (104)-PerCP-Cy5.5 Ab (BD 552950, dilution 1:100) for congenic strain (CD45.1) discrimination. For serial competitive LSK repopulation assays[58], LSK cells (5 × 10$^4$) isolated from WT or KO mice (CD45.2) were mixed with an equal number of LSK cells isolated from competitor mice (WT in CD45.1). The cell mixture was then intravenously injected into lethally irradiated (10 Gy) recipient mice (WT in CD45.1). Secondary transplantation was performed at 12 weeks after primary engraftment. LSK cells (1 × 10$^5$) harvested from primary transplants were intravenously injected into lethally irradiated (10 Gy) 8-week-old recipient mice (WT in CD45.1). The ratio of CD45.1 to CD45.2 positive cells in the BM, PB, thymus, and spleen of recipient mice was measured by flow cytometry.

**In vivo homing assay.** For in vivo homing assay[64], recipient mice were lethally irradiated (10 Gy) 24 h before LSK transplantation. Freshly isolated LSK cells from

8-week-old mice were labeled with 1 μM CFSE (Thermo Fisher Scientific C34554) per the manufacturer's instructions. CFSE-labeled LSK cells (2 × 10$^5$) were then resuspended in PBS and intravenously injected into recipient mice. Sixteen hours after injection, mice were sacrificed, and the frequencies of CFSE$^+$ cells in the BM, PB, and spleen were determined by flow cytometry. To determine the frequency of clonogenic progenitors, mononuclear cells from the BM, PB, and spleen were subjected to CFU assays as described above. To block the interaction between VLA-4 and VCAM-1, CFSE-labeled LSK cells were incubated with an anti-VCAM-1 Ab (2 mg kg$^{-1}$) or an isotype control, and the cells were intravenously injected into recipients 1 h before transplantation.

**Trans-stromal migration and adhesion assay.** For the trans-stromal migration assay[65], LSK cells (1 × 10$^5$ cells chamber$^{-1}$) were loaded into the top chamber of a 24-well transwell plate containing a monolayer of BMSCs. Recombinant mouse CXCL12 (100 ng mL$^{-1}$; R&D Systems P40224) was loaded into the bottom chamber. The plate was then incubated at 37 °C in 5% $CO_2$ for 16 h. After incubation, the number of migrating LSK cells in the bottom chamber was calculated. To perform the adhesion assay, CFSE-labeled LSK cells (5 × 10$^4$) were co-cultured with BMSCs (2 × 10$^5$) in 24-well cell culture plates. Three h after co-culture, nonadherent LSK cells were removed via three gentle washes with PBS, and adherent LSK cells were harvested. The frequency of CFSE$^+$ LSK cells was determined by flow cytometry. To block the VLA-4 and VCAM-1 interaction, LSK cells were preincubated with anti-VLA-4 (10 μg mL$^{-1}$; Merck CBL1304) Ab or an isotype control (10 μg ml$^{-1}$; BD Biosciences 559478) for 30 min, and BMSCs were preincubated with anti-VCAM-1 Ab (10 μg mL$^{-1}$) or an isotype control for 30 min.

**mRNA-seq data analysis.** Total RNAs from WT and KO BM cells were extracted using TRIzol (Thermo Fisher Scientific 15596026) according to the manufacturer's protocol. Subsequently, 1 μg of total RNA was used to construct cDNA libraries using TruSeq RNA library kit (Illumina, RS-122-2001) according to the manufacturer's protocol. The size distribution and quality of cDNA libraries were monitored by Agilent 2100 Bioanalyzer (Agilent Technologies) and quantitative PCR using Kapa Library Quantification Kit (Kapa Biosystems KK4824). After the quality check, cDNA libraries were sequenced using an Illumina HiSeq4000 sequencer (Illumina).

After removing low quality and adapter sequences, the raw reads were aligned to the reference genome *Mus musculus* (mm10) (genome assembly information access, GCF_000001635.20) using HISAT v2.0.5[66]. After aligning reads to the genome, StringTie v1.3.3b was used to assemble transcripts and estimate their abundance measured in fragments of exon per kilobase of exon per million fragments mapped (FPKM)[67,68]. False discovery rate (FDR) was controlled by adjusting $p$-value cutoff of 0.05 using Benjamini–Hochberg algorithm[69]. Subsequently, hierarchical clustering analysis was performed to analyze differentially expressed gene (DEG) set using complete linkage and Euclidean distance as a measure of similarity. Gene set enrichment analysis including functional annotation and pathway analyses were performed based on Gene Ontology (www.geneontology.org/) and KEGG pathway (http://www.genome.jp/kegg/pathway.html). An overview of the gene expression data was deposited at NCBI's Gene Expression Omnibus (GEO). It is accessible through GEO series accession number of GSE128705.

**Real-time quantitative RT-PCR analysis.** Total RNA was purified from BMSCs using Trizol reagent (Thermo Fisher Scientific 15596026) according to the manufacturer's instructions; it was then reverse transcribed into cDNA using a RevertAid First Strand cDNA Synthesis Kit (Thermo Fisher Scientific K1621). After cDNA synthesis, quantitative PCR (qPCR) was conducted to measure mRNA levels of mouse *Alcam, Angpt1, Cdh2, Ctsk, Cxcl12, Fn1, Kitl, Icam1, p16$^{INK4a}$, p19$^{ARF}$, Selp, Vcam1,* and *Gapdh* using a 7500 Real-time PCR system (Thermo Fisher Scientific). *Gapdh* was used as an internal control to quantify the levels of other mRNA transcripts. The primer pairs used for qRT-PCR are listed in Supplementary Table 1.

**Inhibition of *Phc2* expression by shRNA.** Lentiviral plasmids expressing shRNAs directed against several distinct regions of *Phc2* (SHCLNG-NM_018774) and control shRNA plasmids (SHC001) were purchased from Sigma-Aldrich. OP9 cells (mouse BMSC line, ATCC CRL-2749) were transfected with each shRNA-containing plasmid using Lipofectamine 2000 transfection reagent (Thermo Fisher Scientific 11668019) according to the manufacturer's instructions. After transfection, inhibition of *Phc2* expression was determined by qRT-PCR and immunoblotting. Primer sequences for *Phc2* for qRT-PCR are listed in Supplementary Table 1.

**ChIP assay.** The lysates of BM cells (2 × 10$^6$) from 8-week-old mice were prepared for the ChIP assay[70]. BM cell lysates were sonicated and immunoprecipitated with the anti-Bmi1 (Santa Cruz Biotechnology, Inc. sc-10745, 2 μg sample$^{-1}$), anti-Cbx7 (Santa Cruz Biotechnology, Inc. sc-70232, 2 μg sample$^{-1}$), anti-H2AK119ub (Cell Signaling Technology 8240, 2 μg sample$^{-1}$), anti-H3K9ac (Merck 06-942, 2 μg sample$^{-1}$), anti-H3K4me3 (Abcam ab12209, 2 μg sample$^{-1}$), anti-H3K27me3 (Merck 07-449, 2 μg sample$^{-1}$), anti-Phc2 (Santa Cruz Biotechnology, Inc. sc-

160664, 2 μg sample$^{-1}$), and anti-Ring1b (Cell Signaling Technology 5694, 2 μg sample$^{-1}$) Abs. Additionally, GSK126-treated OP9 cell lysates were sonicated and immunoprecipitated with the anti-Bmi1, anti-Phc2, and anti-Ring1b Abs. After immunoprecipitation, immune complexes were collected with protein A agarose (Merck GE17-0963-03) and extracted with an extraction buffer (1% SDS, 0.1 M NaHCO$_2$). DNA cross-links were reversed by heating to 65 °C for 8 h. DNA was extracted with phenol/chloroform and precipitated with ethanol. DNA isolated from an aliquot of total nuclear extract was used as the loading control for PCR (input control). qPCR was performed as described above. Data are presented after normalizing each immunoprecipitated DNA Ct value to 10% of the input DNA Ct value.

**In vivo administration of GSK126**. WT mice were intraperitoneally injected with 50 or 100 mg kg$^{-1}$ of DMSO-dissolved GSK126. Mice were sacrificed at 16 h after GSK126 injection and the expression of VCAM-1 from BMSCs of sacrificed mice were analyzed by immunoblotting and flow cytometry. In addition, trans-stromal migration and adhesion assays using LSK cells and BMSCs of sacrificed mice were performed as described above.

**Statistical analysis**. Mean values among more than three experimental groups were compared by one-way ANOVA with Tukey HSD analysis and comparison between two groups was analyzed by two-tailed Student's *t*-test using SPSS statistics software, ver. 24.0 (SPSS Inc). All data are presented as the mean ± standard error of mean (SEM). In each graph, significant differences as determined by *p*-values less than 0.05 and 0.01 are indicated by asterisks (* and **, respectively).

**Reporting summary**. Further information on research design is available in the Nature Research Reporting Summary linked to this article.

## Data availability

The raw mRNA-seq data were deposited into the Sequence Read Archive (SRA) of National Center for Biotechnology Information (NCBI) with the following accession numbers: GSM3683312, GSM3683313, GSM3683314, GSM3683315, GSM3683316, and GSM3683317. An overview of the gene expression data was deposited at NCBI's Gene Expression Omnibus (GEO). It is accessible through GEO series accession number of GSE128705. Source data underlying Figs. 1–8, Tables 1–2, Supplementary Figs. 1–7, and Supplementary Figs. 10–18 are provided as a Source Data file. All other data that support the findings of this study are available from the corresponding author upon reasonable request.

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

## Acknowledgements

We thank Amgen Inc. (Thousand Oaks, CA, USA) for providing G-CSF. We also thank Eun-Kyeong Jo (Chungnam National University School of Medicine), Kyung-Mi Lee (Korea University College of Medicine), Seungkwon You (Korea University School of Life Sciences and Biotechnology), and Seok-Ho Hong (Kangwon National University College of Medicine) for their critical reading of our manuscript or helpful suggestions. This research was supported by the Bio & Medical Technology Development Program of the National Research Foundation (NRF) funded by the Ministry of Science & ICT (2017M3A9C8060392), and an intramural research program funded by Korea University grant (2016).

## Author contributions

T.C. conceived, designed, and supervised this study. J.B., S.-P.C., K.I., J.Y.L., S.-W.P., C.-Y.C., J.H., S.-H.K., H.-H.L., K.P. and S.J.L. performed experiments. J.B., S.-P.C., C.-Y.C., C.-G.P. and T.C. analyzed and interpreted data. J.B., S.-P.C., H.Y.J., C.-G.P., H.K., Y.S.L. and T.C. discussed, wrote and edited the manuscript. All authors reviewed and approved the manuscript.

## Additional information

**Competing interests:** The authors declare no competing interests.

