## [Peer Review File · Nature Communications]

Reviewers' comments:

Reviewer #1 (Remarks to the Author):

In this study, the authors show that Phc2, a component of the PRC1 complex, regulates HSPC trafficking by modulating VCAM1 expression on bone marrow stromal cells. This is a novel observation that provides important new insight into how epigenetic modifiers may regulate hematopoiesis in a non-cell extrinsic fashion. Although the data are in general convincing, there are several major concerns that need to be addressed.

Major Concerns

1. In assessing HPSC mobilization, it is important to compare HSPC numbers at baseline with those after mobilization. Thus, the baseline data for BM, blood, and spleen (currently in Suppl Fig 1) should be incorporated into Figure 2. With the decreased baseline HSPCs in the blood and spleen of Phc2 mice, AMD3100-induced mobilization may not be significant.
2. A key point in this study, is that Phc2 works in a non-hematopoietic cell intrinsic fashion to regulated HSPC trafficking. The most convincing evidence for this is the bone marrow chimera analysis. In this regard, it is disappointing that only total nucleated cell numbers were measured in hematopoietic organs. HPSCs should be quantified at baseline and after mobilization with G-CSF in the bone marrow chimeras.
3. The bone sections showing co-staining of VCAM1 with the different stromal cell populations is not convincing (Figure 5, Supplemental Figure 7). No clear vascular pattern is seen with the CD31 staining. OPN appears to stain megakaryocytes and not osteoblasts. It is not clear what cells are being stained with SSEA4. Better quality images are needed.
4. Papayannopoulou et al (Blood, 1998) showed that anti-VCAM-1 induces HPSC mobilization. Yet, no clear mobilization effect was observed in Figure 7 (perhaps because of G-CSF or 5-FU treatment). In any case, the effect of anti-VCAM-1 alone on HPSC numbers in hematopoietic tissues should be shown.

Minor Concerns

1. In figure 2h, it would be better to show the CFU-C data on day 8 after 5-FU in addition to data on day 16.
2. In supplementary figure 4, it is standard to show the contribution of donor cells to each mature hematopoietic lineage, in addition to total donor cell chimerism.
3. The number of MSCs identified in the bone marrow is much higher than prior reports. Indeed, the gating strategy (Suppl. Fig. 5) used to identify MSCs is not standard.
4. How were the data in Figure 5f normalized? The authors should comment on the lack of change in basal VCAM1 expression. How does this related to the observed in vivo phenotypes.
5. There may be mistake in the legend for Figure 7a/b. The legend states that WT vs KO HSPC migration through KO stromal cells was assessed. It seems more likely that WT HPSC migration through KO or WT stromal cells were studied.
6. In Figure 7c, please clarify what the genotype of recipient mice was used.

Reviewer #2 (Remarks to the Author):

General Comments

In this study by Joonbeom et. al., the authors have demonstrated that Phc2, a component of PRC1 complex, regulates mobilization of HSPCs by repressing Vcam1 expression in bone marrow stromal

cells (BMSCs). Phc2 deficiency apparently caused reduced mobilization of HSPCs from bone marrow to PB and spleen. Mechanistically, they suggest that the Phc2 complex binds to specific sites in the Vcam1 transcription unit in BMSCs. Loss of function of Phc2 changed histone modification associated with Vcam1 and thus enhanced expression of Vcam1 which was normally repressed by the PRC1 complex. Neutralizing VCAM-1 restored HSPC mobilization defect caused by loss of Phc2.

While of interest, there are concerns that need to be addressed.

Specific Comments

1. As an important component of polycomb repressor complex 1 (PRC1), it is surprising that Phc2 deficiency did not result in any defect of HSPC maintenance and self-renewal since Bmi-1, another component of PRC1, is well-known for its role in HSC maintenance; Ezh2, a component of PRC2, is also required for HSC maintenance. The authors need to comment on this. Can the authors give some genetic information of this Phc2 KO mouse? Did they check whether the function of Phc2 was indeed lost? Or is it just a weak allele?
2. Supplementary Fig.2: the frequencies and absolute numbers of LSK cells in the PB and spleen were significantly lower in Phc2 KO mice. This is an important evidence of reduced mobilization of HSPCs, but this needs to be further proved by in vivo transplantation experiments to assess numbers of functional hematopoietic stem cells (via competitive repopulating and limiting dilution analysis to calculate competitive repopulating units). It is better to put these data in major figures.
3. Fig.3h and Supplementary Fig. 4: In order to prove Phc2 KO HSPCs had no intrinsic defect, the authors need to do the transplantation experiments using sorted SLAMF6⁺ LSK or at least LSK cells.
4. Phc2 regulates expression of Vcam-1 in BMSCs. As the authors mentioned, there are three major BMSCs, including endothelial cells, mesenchymal stem cells and osteoblasts. The authors need to figure out in which population of cells repression of Vcam-1 by Phc2 mainly happens.
5. Ezh2 inhibition can decrease binding of Phc2, Bmi1 and Ring1b to the transcription unit of Vcam-1. Does this mean that the PRC2 complex functions upstream of PRC1 including Phc2 in repression of Vcam-1 expression? Again, back to first point, in Bmi1 or Ezh2 KO mice, did the authors observe a similar phenotype of HSPC mobilization in Phc2 KO mice? If not, how do they explain this difference?
6. Other studies have already shown that the VLA-2/VCAM-1 pathway is the downstream target of Phc2, and the authors just proved this in BMSCs. Since PRC1 is a classical epigenetic repressor which is involved in regulation of many genes, the authors need to perform an RNA-seq analysis to see whether other potential targeted genes are involved in Phc2 mediated HSPC mobilization.
7. The authors showed that treatment with an Ezh2 inhibitor, GSK126, resulted in upregulation of Vcam1 expression in OP9 cells. This should be tested in vivo to see whether administration of GSK126 would mimic the phenotype of Phc2 deficiency in Vcam1 upregulation and HSPC mobilization. This could have potential clinical application.
8. Page 7, last 3 lines: Clarify how you know that this is decreased migration.
9. Page 8, line 6: Data in Figure 1g should be shown as absolute numbers of colonies in bone marrow (e.g. femur), and to add to this what were the absolute numbers of colony forming cells of each type (CFU-GM, BFU-E and CFU-GEMM) in spleen and thymus?

10. Supplementary Figure 1: I assume that CFU-C refers to CFU-GM unless this is for all colonies so be consistent in nomenclature between Figures (e.g. Fig 1g), and why is this information not available for each progenitor cell type (CFU-GM, BFU-E and CFU-GEMM) also?

11. Figure 2d and the CFU-C/ml PB in Fig. 2e: These decreases, although significant are very modest with AMD3100. Since AMD3100 synergizes with G-CSF to mobilize stem and progenitor cells it would be important to assess this in context of mobilization with G-CSF and AMD3100 as has been reported by others.

Minor Comments

1. In Page 9 Line 137, "WT mice were significantly..." should be "KO mice were significantly...". In addition, statistics are missing in Fig5f and g.

2. Page 9, line 137: Shouldn't WT mice should be Phc2 KO mice?

Reviewer #3 (Remarks to the Author):

Mobilized peripheral blood stem cells (PBSCs) have evolved as the main cellular source for HSPC transplants in patients. While the impact of bone marrow niche in the hematopoiesis is well recognized, the mechanisms underlying the interactions between HSPCs and niche compartment remains largely unknown. In the current study, Bae and colleagues aim to investigate the role of Phc2, an epigenetic regulator, in HSPC mobilization. Using a mouse model of Phc2, the authors showed that genetic deletion of Phc2 resulted in a defect in HSPC mobilization through H3K27me3- and H2AK119ub-mediated derepression of Vcam1 in the bone marrow stromal cells (BMSCs). This study provides scientific insights for a cell-extrinsic effect of Phc2 in the regulation of HSPC mobilization. While this study has its novelty, there are multiple concerns. For the mobilization assays, detailed characterization for the changes in different lineages is missing. Some of the conclusion is over stated.

Major concerns:

1. The rationale for studying the Phc2 in HSPC mobilization is not clear.
2. In the introduction, the authors mentioned that "several BM niche proteins have been identified as key signaling components to control the maturation of HSPCs". Some of the proteins in the niche critical for HSPC functions should be listed.
3. The specific types of the CFUs should be clarified.
4. What is the impact of the Phc2 deletion on the myeloid lineages in different tissue organs?
5. Figure 4, in an in vivo homing assay, the authors measured the number of CFSE+ cells homing to the BM. Since the CFSE+ cells may be a mixed population, more detailed characterization is needed to determine the specific lineage populations in the CFSE+ cell population.
6. In Figure 4c, the authors assessed the frequency of CFU-Cs in the BM, PB and spleen of the recipient mice. Again, specific types of the CFUs should be characterized.
7. In Figure 7 c,d, the authors quantified CFU-C and CFSE+ cells in an in vivo homing assay following the blockage of the interaction between VCAM-1 and VLA-4. The authors claimed that "blocking the interaction between VCAM-1 and VLA-4 could restore LSK cell mobilization from the KO BM into the PB and spleen". The frequencies of specific HSC/HPC populations by flow cytometric analyses should be provided.
8. While the authors provided CHIP-qPCR data showing the Phc2 binding sites on the Vcam1 gene,

ChIP-seq may provide strong evidence for the role of Phc2 on H2AK119ub and H3K27 methylation-mediated gene regulation.

9. All the statistical data is based on the two-tailed Student's t test. However, since many of the study groups were four, statistical analyses with ANOVA would be more appropriate.

Minor concerns

1. Supplementary Fig. 3 in line 155 on page 10 should be Supplementary Fig. 4.

2. In Figure 5, the authors showed a higher level of Vcam1 in Phc2^{-/-} BMSCs due to a reduction of H3K27me3 and H2AK119ub levels. However, the authors stated that "Phc2 can directly repress Vcam1 expression through histone modifications". The "directly" may not be the appropriate word.

3. Please do spelling and grammar check for the paper.

Response to Reviewers

We appreciate the reviewers for their helpful and constructive comments. We have carefully read through the reviewer's comments and changed the manuscript according to the reviewer's suggestions.

We hope that this revised manuscript strengthens the main conclusion (the extrinsic rule of epigenetic factor on bone marrow niche as a novel important regulator of hematopoietic stem/progenitor cell mobilization) of the original manuscript.

For reviewer's convenience, we prepared two different versions of revised manuscript. One is an unmarked version and another is a marked version. In marked version, all changes in text are marked with yellow highlights.

Please find a point-by-point responses to each of the reviewers' comments/questions along with additional data to support our findings. All corrections are detailed below.

Specific remarks

Reviewer #1

Major Concerns

1) In assessing HPSC mobilization, it is important to compare HSPC numbers at baseline with those after mobilization. Thus, the baseline data for BM, blood, and spleen (currently in Suppl Fig 1) should be incorporated into Figure 2. With the decreased baseline HSPCs in the blood and spleen of Phc2 mice, AMD3100-induced mobilization may not be significant.

: Thank you for your constructive comment. According to your comment, we incorporate supplementary Figure 1 into Figure 2c, 2e and 2h in the revised manuscript.

When we used an AMD3100 as a mobilizing agent, we still observed that Phc2 KO HSPC migration are significantly decreased compared with WT HSPC migration. However, the difference is relatively modest when we compared with results from G-CSF and 5-FU treatment.

According to you and reviewer #2's suggestion, we treated G-CSF and AMD3100 since previous observation reported that G-CSF and AMD3100 synergize the mobilization of HSPCs¹⁷. Results clearly indicated that Phc2 KO HSPC migration are significantly decreased compared with WT HSPC migration (Supplementary Figure 17 in the revised manuscript).

According to these changes, we added one reference and changed the revised version of manuscript (line 279 - 280 in page 19, lines 493 - 497 in page 33).

References

17. Broxmeyer HE, *et al.* Rapid mobilization of murine and human hematopoietic stem and progenitor cells with AMD3100, a CXCR4 antagonist. *J. Exp. Med.* **201**, 1307-1318 (2005).
60. Ramirez, P., *et al.* BIO5192, a small molecule inhibitor of VLA-4, mobilizes hematopoietic stem and progenitor cells. *Blood* **114**, 1340-1343 (2009).

2) A key point in this study, is that Phc2 works in a non-hematopoietic cell intrinsic fashion to regulated HSPC trafficking. The most convincing evidence for this is the bone marrow chimera analysis. In this regard, it is disappointing that only total nucleated cell numbers were measured in hematopoietic organs. HSPCs should be quantified at baseline and after mobilization with G-CSF in the bone marrow chimeras.

: Thank you for your constructive comment. We absolutely agreed with your opinion. According to your comment, we assessed the significance of VCAM-1 blockage when the migration defect of HSPCs are driven by extrinsic microenvironment which mimics the phenotype associated with Phc2 KO mice.

As a result, the administration of the neutralizing VCAM-1 Ab in combination with G-CSF in KO recipient mice completely reconstituted WT donor HSPC mobilization from BM into periphery, the same degree to WT recipient mice (Supplementary Figure 18 in the revised manuscript). Consistent to the HSPC mobilization assay in KO mice, WT cells reconstituted in irradiated KO recipient cells were only partially rescued the mobilization defects when the single G-CSF agent was treated (Supplementary Figure 18 in the revised manuscript).

According to these changes, we changed the revised version of manuscript (line 281 - 292 in page 19).

3) The bone sections showing co-staining of VCAM1 with the different stromal cell populations is not convincing (Figure 5, Supplemental Figure 7). No clear vascular pattern is seen with the CD31 staining. OPN appears to stain megakaryocytes and not osteoblasts. It is not clear what cells are being stained with SSEA4. Better quality images are needed.

: Thank you for your constructive comment. According to your comment, we did additional experiments and replaced figures (Figure 5, Supplemental Figure 10) in revised version of the manuscript.

According to these changes, we changed the revised version of manuscript (line 453 - 456 in page 31).

4) Papayannopoulou et al (Blood, 1998) showed that anti-VCAM-1 induces HPSC mobilization. Yet, no clear mobilization effect was observed in Figure 7 (perhaps because of G-CSF or 5-FU treatment). In any case, the effect of anti-VCAM-1 alone on HPSC numbers in hematopoietic tissues should be shown.

: Thank you for your constructive comment. We absolutely agreed with your opinion. According to your comment, we did additional experiments to see the effect of anti-VCAM-1 alone on HPSC numbers in hematopoietic tissues. Indeed, treatment of the neutralizing VCAM-1 Ab could rescue steady-state HSPC mobilization in KO mice (Figure 7e-g in the revised manuscript).

According to these changes, we changed the revised version of manuscript (line 265 - 268 in page 18, line 511 in page 34 – line 516 in page 35, line 919 – 923 in page 63).

Minor Concerns

1) In figure 2h, it would be better to show the CFU-C data on day 8 after 5-FU in addition to data on day 16.

: Thank you for your constructive comment. According to your comment, we incorporated CFU-C data on day 8 after 5-FU (Figure 2h in the revised manuscript).

2) In supplementary figure 4, it is standard to show the contribution of donor cells to each mature hematopoietic lineage, in addition to total donor cell chimerism.

: Thank you for your constructive comment. Actually, reviewer #2 asked us to do LSK transplantation experiments for all of BM repopulation assays including serial competitive assays. According to reviewer #2's opinion, we did several LSK transplantation experiments. When we did serial competitive assays with LSK transplantation, we monitored the contribution of donor cells to each mature hematopoietic lineage according to your opinion (Supplementary Figure 4 in the revised manuscript).

3) The number of MSCs identified in the bone marrow is much higher than prior reports. Indeed, the gating strategy (Suppl. Fig. 5) used to identify MSCs is not standard.

: Thank you for your constructive comment. According to your comment, we re-evaluated the number of MSCs identified in the bone marrow using new gating strategy (Supplementary Figure 7 in the revised manuscript).

4) How were the data in Figure 5f normalized? The authors should comment on the lack of change in basal VCAM1 expression. How does this related to the observed in vivo phenotypes.

: We acknowledge your comment. Figure 5f indicated that the mRNA expression of *Vcam1* reversely correlated with *Phc2* expression. Supporting this notion, Ezh inhibitor, GSK126 treatment increases expression of VCAM-1 mRNA and protein levels (Figure 6d and 6e).

We are sorry to confuse you. Actually, the level of *Vcam1* mRNA transcripts was normalized with that of *Gapdh* and relative fold change of each *Vcam1* mRNA expression from Control or *shPhc2* was calculated relative to basal level of that in unstimulated Control or *shPhc2*, respectively. Therefore, Values "1" in y-axis in Figure 5f and 5g mean basal levels of VCAM-1 mRNA and protein expression in unstimulated Control or *shPhc2*, respectively.

As mentioned in discussion section, the BMSC is the only cell type that constitutively expresses VCAM-1 at physiological levels in steady-state conditions. Therefore, vascular endothelial cells and other immune cell populations express minimally detectable levels of VCAM-1³³⁻³⁸ (line 321 – 324 in the revised manuscript). OP9 cells express minimally detectable levels of VCAM-1 which is inducible by TNF-alpha as shown in Figure 5f and 5g.

According to your comment, we changed the label of y-axis to “relative fold change of Vcam1 expression” in Figure 5f and 5g, and revised the manuscript (line 932 -933 in page 64).

5) There may be mistake in the legend for Figure 7a/b. The legend states that WT vs KO HSPC migration through KO stromal cells was assessed. It seems more likely that WT HPSC migration through KO or WT stromal cells were studied.

: Thank you very much for your kindness. We fixed sentence regarding the legend for Figure 7a/b in the revised manuscript (line 951 – 952 in page 65).

6) In Figure 7c, please clarify what the genotype of recipient mice was used.

: We acknowledge your comment. According to reviewer's comment, we clarify what the genotype of recipient mice was used in the revised manuscript (line 954 – 956 in page 65).

Reviewer #2

Specific Comments

1) As an important component of polycomb repressor complex 1 (PRC1), it is surprising that Phc2 deficiency did not result in any defect of HSPC maintenance and self-renewal since Bmi-1, another component of PRC1, is well-known for its role in HSC maintenance; Ezh2, a component of PRC2, is also required for HSC maintenance. The authors need to comment on this. Can the authors give some genetic information of this Phc2 KO mouse? Did they check whether the function of Phc2 was indeed lost? Or is it just a weak allele?

: Thank you very much for your constructive comment. Your comment inspires us to emphasize “the heterogeneity of PRC1 or PRC2 complex by paralog of each protein subunit of PcG

complex” in the “discussion” section of revised manuscript. We also did additional experiment to support this idea. Combined results will make our result or discussion more meaningful and make the reader more interested about the functional heterogeneity of individual PcG subunit protein.

Phc2 KO mice were C57BL/6 background and originally described in 2005¹⁴.

References

14. Isono, K., et al. Mammalian polyhomeotic homologues Phc2 and Phc1 act in synergy to mediate polycomb repression of Hox genes. *Mol. Cell Biol.* **25**, 6694-6706 (2005).

Like deficiencies in other epigenetic regulators, a single PcG gene deficiency in mice mostly causes early embryonic lethality or a limited life span with severe developmental defects, including an intrinsic defect in HSCs^{8, 46, 47}. This characteristic hinders the search for additional roles of polycomb proteins in the regulation of immune phenomena aside from their known functions in HSC differentiation.

Similar to other members of the PcG protein family, Phc2 is expressed in various tissues and cells and regulates the transcription of *Cdkn2a* (*p16^{INK4a}* and *p19^{ARF}*) and *hox* genes through direct associations with chromatin^{8,14}. However, Phc2-deficient mice are fertile and exhibit less severe defects associated with the abnormal axial skeleton and Peyer's patch^{14, 48}. Moreover, the systemic immunodeficiency observed in Phc2-deficient mice is not caused by an intrinsic defect in HSC function, but instead, the Phc2 deficiency causes a defect in HSPC mobilization by enhancing the expression of VCAM-1 in BMSCs. This may suggest that most of the Phc2 functions in mammals are dispensable or can be replaced by other components of the canonical PRC1 due to a structural heterogeneity of PRC1 or functional redundancy of PcG proteins.

In fact, each of the mammalian PRC2 or PRC1 subunits has several paralogs⁸⁻¹³. For example, Phc2 has two paralogs, Phc1 and Phc3⁸⁻¹³. Previous observation revealed that Phc1 deficiency causes a defect in HSC's self-renewal activity^{49, 50}. Therefore, Phc1, but not phc2, might be critical for HSC maintenance and function. Consistent with this idea, the expression and epigenetic pattern of *Cdkn2a* (*p16^{INK4a}* and *p19^{ARF}*) in KO BM cells showed no significant differences compared to those in WT BM cells (Supplementary Fig. 19). Another example of

nonredundant function by each paralog within PRC2 or PRC1 components might be found in Ezh proteins or Cbx proteins, respectively⁵¹⁻⁵⁵. Among Ezh proteins, Ezh1 is critical for self-renewal activity of HSCs, whereas Ezh2 is dispensable for proper HSC function^{51, 52}. Also, Cbx7 is the only paralog which is necessary for self-renewal activity of HSC among Cbx proteins^{53, 54}.

Supplementary Figure 19. The expression and epigenetic pattern of *Cdkn2a* in KO BM cells show no significant differences compared to those in WT BM cells. **a** mRNA expression patterns of *p16^{Ink4a}* (left) and *p19^{ARF}* (right) in WT or KO BM cells were accessed by qRT-PCR. $n = 3$. **b** ChIP was performed with WT or KO BM cell lysates using indicated Abs or a control Ab (Control). DNA recovered from ChIP was then analyzed by qPCR. $n = 5$. 1st exon, *P16^{Ink4a}* first exon; 2nd exon, *P16^{Ink4a}/p19^{ARF}* common second exon. The primer pairs used for qRT-PCR are listed in Supplementary Table 1.

According to these changes, we added new references in the revised manuscripts and revised the manuscript (line 356 in page 24 – line 367 in page 25).

New references

49. Ohta, H., et al. Polycomb group gene *rae28* is required for sustaining activity of hematopoietic stem cells. *J. Exp. Med.* **195**, 759-770 (2002).
50. Kim, J. Y., et al. Defective long-term repopulating ability in hematopoietic stem cells lacking the polycomb-group gene *rae28*. *Eur. J. Haematol.* **73**, 75-84 (2004).
51. Hidalgo, I., et al. Ezh1 is required for hematopoietic stem cell maintenance and prevents senescence-like cell cycle arrest. *Cell Stem Cell* **11**, 649-662 (2012).

52. Xie, H., et al. Polycomb repressive complex 2 regulates normal hematopoietic stem cell function in a developmental-stage-specific manner. *Cell Stem Cell* **14**, 68-80 (2014).
53. van Den Boom, V., et al. Nonredundant and locus-specific gene repression functions of PRC1 paralog family members in human hematopoietic stem/progenitor cells. *Blood* **121**, 2452-2461 (2013).
54. Klauke, K., et al. Polycomb Cbx family members mediate the balance between haematopoietic stem cell self-renewal and differentiation. *Nat. Cell Biol.* **15**, 353-362 (2013).

2) Supplementary Fig.2: the frequencies and absolute numbers of LSK cells in the PB and spleen were significantly lower in Phc2 KO mice. This is an important evidence of reduced mobilization of HSPCs, but this needs to be further proved by in vivo transplantation experiments to assess numbers of functional hematopoietic stem cells (via competitive repopulating and limiting dilution analysis to calculate competitive repopulating units). It is better to put these data in major figures.

: Thank you for your constructive comment. According to your comment, we did additional experiments to assess numbers of functional HSCs more carefully.

Following is the summary of additional experiments:

1. We did LSK transplantation experiments for BM chimera and did serial competitive repopulation assays using LSK cells (Figure 3 and Supplementary Figure 4).
2. We also did short-term migration assay using LSK cells (Figure 4 and Figure 7).
3. In addition, we measured the frequencies of donor-derived clonogenic progenitors (CFU-GM, BFU-E, CFU-GEMM) homing to the bone marrow, peripheral blood and spleen. (Supplementary Figure 5 and Supplementary Figure 14).
4. Finally, we assessed the significance of VCAM-1 blockage when the migration defect of HSPCs are driven by extrinsic microenvironment which mimics the phenotype associated

with Phc2 KO mice. As a result, the administration of the neutralizing VCAM-1 Ab in combination with G-CSF in KO recipient mice completely reconstituted WT donor HSPC mobilization from BM into periphery, the same degree to WT recipient mice (Supplementary Figure 18 in the revised manuscript). Consistent to the HSPC mobilization assay in KO mice, WT cells reconstituted in irradiated KO recipient cells were only partially rescued the mobilization defects when the single G-CSF agent was treated (Supplementary Figure 18 in the revised manuscript).

Supplementary Figure 18. G-CSF induced HSPC mobilization assay in BM chimeric mice. LSK cells (1×10^5) from WT mice (CD45.1) were injected intravenously into lethally irradiated WT or KO CD45.2 recipient mice. Twelve weeks later, G-CSF or vehicle was administered by daily subcutaneous injection to chimeric mice at a dose of 250 $\mu\text{g}/\text{kg}/\text{day}$ for 5 days. To block the VLA-4 and VCAM-1 interaction, anti-VCAM-1Ab or the respective isotype control (2 $\text{mg}/\text{kg}/\text{day}$) was administered intravenously every day. **a** Schematic representation of LSK transplantation and G-CSF induced mobilization assay. **b** Comparison of WBC counts between BM chimeric mice on day 5 after G-CSF and/or anti-VCAM1 Ab treatment. **c** Representative image of spleens from BM chimeric mice (*left*) and absolute numbers of splenocytes for BM chimeric mice (*right*) on day 5 after G-CSF and/or anti-VCAM1 Ab treatment. **c** CFU-C in the BM (femur, *left*), PB (*middle*), and spleen (SP, *right*) of BM chimeric mice on day 5 after G-CSF and anti-VCAM1 Ab treatment. WT, WT recipient mice; KO, KO recipient mice. "-", treatment with vehicle or isotype control Ab; "+", treatment with G-CSF or anti-VCAM-1 Ab. All data are presented as the mean \pm SEM. * $P < 0.05$; ** $P < 0.01$.

According to these changes, we revised the manuscript (line 166, 172, 173 and 174 in page 12, line 182, 184, 185, 189, 192 and 194 in page 13, line 281 – 292 in page 19, line 518 in page 35 – line 548 in page 37, line 891 page 61 – line 905 in page 62).

3) Fig.3h and Supplementary Fig. 4: In order to prove Phc2 KO HSPCs had no intrinsic defect, the authors need do the transplantation experiments using sorted SLAM LSK or at least LSK cells.

: Thank you for your constructive comment. As mentioned in question #2, we did LSK transplantation experiments for BM chimera and did serial competitive repopulation assays using LSK cells (Figure 3 and Supplementary Figure 4).

4) Phc2 regulates expression of Vcam-1 in BMSCs. As the authors mentioned, there are three major BMSCs, including endothelial cells, mesenchymal stem cells and osteoblasts. The authors need to figure out in which population of cells repression of Vcam-1 by Phc2 mainly happens.

: Thank you for your constructive comment. According to your comment, we did additional experiments to determine which cell population of BMSCs requires repression of Vcam-1. Briefly, endothelial cells (Endo), osteoblasts (OB) or mesenchymal stem cells (MSC) in WT or KO BM were isolated by FACS sorting. Then, trans-stromal migration assays were performed to measure the capacity of WT LSK cells to transmigrate across each subpopulation of WT or KO BMSCs. As a result, all three subpopulations of BMSCs are responsible for the repression of VCAM-1 (Supplementary Fig. 6).

5) Ezh2 inhibition can decrease binding of Phc2, Bmi1 and Ring1b to the transcription unit of Vcam-1. Does this mean that the PRC2 complex functions upstream of PRC1 including Phc2 in repression of Vcam-1 expression? Again, back to first point, in Bmi1 or Ezh2 KO mice, did the authors observe a similar phenotype of HSPC mobilization in Phc2 KO mice? If not, how do they explain this difference?

: We acknowledge your comment. This question is related to question #1. As mentioned in question #1, we believed that the phenotypic differences of each PcG subunit genes is related

to the structural heterogeneity of PRC1 or PRC2 complex by paralog of each protein subunit of PcG complex.

We also believed that the PRC2 complex functions upstream of PRC1 including Phc2 in repression of *Vcam-1* expression.

For reader's better understanding about the function of PRC2 and PRC1, we wrote following sentences in "Introduction" section (line 82 -91 in page 6) and "Discussion" section (line 368 in page 25 – line 379 in page 26) of the revised manuscript.

"Introduction" section

Polycomb group (PcG) proteins function as transcriptional repressors of target genes by mainly modulating histone methylation⁸. These PcG proteins can be divided into two functionally distinct complexes: polycomb repressor complex 1 (PRC1) and polycomb repressor complex 2 (PRC2). PRC2 induces the formation of trimethylated histone H3 at lysine 27 (H3K27me3) through methyltransferase activity of Ezh proteins which are component proteins in PRC2 complex⁹. The canonical PRC1 then recognizes and binds to H3K27me3 to sustain the transcriptional repression of a target gene^{8, 10, 11}. Both canonical and noncanonical PRC1 complexes can cause additional transcriptional silencing as E3 ubiquitin ligases by forming monoubiquitylated histone H2A at lysine 119 (H2AK119ub)^{8, 12, 13}.

"Discussion" section

In our molecular model, PRC2 containing Ezh2 with methyltransferase activity initiates H3K27me3 on the *Vcam1* locus in BMSCs because the Ezh2 inhibitor (GSK126) treatment of BMSCs increases the *Vcam1* mRNA levels by negating the recognition of H3K27me3 by the canonical PRC1 complex containing Phc2 on the same locus. The canonical PRC1 then recognizes and binds to H3K27me3 to maintain *Vcam1* gene suppression. The recognition of H3K27me3 on the *Vcam1* locus by the canonical PRC1 is dependent on *Phc2* expression. Consistently, we observed reductions in the binding of other canonical PRC1 components to the *Vcam1* locus when Phc2 was not expressed. Our study also revealed that the H2AK119ub activity of the canonical PRC1 on the *Vcam1* locus was dependent on *Phc2* expression, which

is consistent with a recent observation that the interaction between Bmi1 and Phc2 is critical for the H2AK119ub activity of the canonical PRC1⁵⁵.

6) Other studies have already shown that the VLA-4/VCAM-1 pathway is the downstream target of Phc2, and the authors just proved this in BMSCs. Since PRC1 is a classical epigenetic repressor which is involved in regulation of many genes, the authors need to perform an RNA-seq analysis to see whether other potential targeted genes are involved in Phc2 mediated HSPC mobilization.

: We acknowledge your comment. You may mention the following manuscript.

Huang, T. S., et al. LINC00341 exerts an anti-inflammatory effect on endothelial cells by repressing VCAM1. *Physiol Genomics* **49**, 339-345 (2017).

We carefully read above manuscript. Their work focused on the function of LINC00341, one of the most abundant long noncoding RNAs, as a suppressor of *Vcam1* expression. In agreement with our data, they showed that Ezh2 binds the promoter region of *Vcam1*. However, they did not show any direct evidence indicating the relationship between Ezh2 and VCAM-1 expression. For example, they did not show that the inhibition of Ezh2 upregulates VCAM-1. They did not point out which promoter region is the binding site of PcG proteins, either. Therefore, they never did systemic analyses to define the functional role of PRC2 and PRC1 for repressing *Vcam1*.

As shown in Figure 5, Figure 6, Supplementary Figure 11 and Supplementary 13, our study systemically revealed the contribution of PRC2 and PRC1 for repressing *Vcam1*. We also demonstrated the biological significance of that suppression.

According to your suggestion, we perform an RNA-seq analysis (Supplementary Figure 9). Results were consistent with real-time qPCR result shown in Figure 5a.

Supplementary Figure 9. Overview of RNA seq and differential expression analysis in WT and KO BM cells. **a** Hierarchical clustering analysis of differentially expressed genes (DEGs) between WT and KO BM cells. 'Red' indicates relatively higher expression and 'green' indicates relatively lower expression. **b, c** GO functional enrichment analysis according to biological process (**b**) and molecular function (**c**). X axis represents DEG count while Y axis represents GO terms. **d** mRNA expression patterns of genes related to HSPC mobilization in WT and KO BM cells. '+' value means increased mRNA expression of indicated gene in KO BM cells compared to that in WT BM cells, whereas '-' value means decreased mRNA expression of indicated gene in KO BM cells compared to that in WT BM cells.

According to these changes, we revised the manuscript (line 209 in page 14, line 566 in page 38 – 593 in page 39).

7) The authors showed that treatment with an Ezh2 inhibitor, GSK126, resulted in upregulation of Vcam1 expression in OP9 cells. This should be tested in vivo to see whether administration of GSK126 would mimic the phenotype of Phc2 deficiency in Vcam1 upregulation and HSPC mobilization. This could have potential clinical application.

: We acknowledge your comment. According to your suggestion, we asked whether GSK126 treatment could mimic Phc2 deficiency *in vivo*. Different dose of GSK126 was administered intraperitoneally into WT mice. Sixteen h after GSK injection, BMSCs were isolated and analyzed for VCAM-1 expression. We also measured the capacity of WT LSK cells to transmigrate across their GSK126-treated BMSCs using a trans-stromal migration assay.

Results clearly showed that GSK126 treatment increased both VCAM-1 expression in BMSCs and adherence of LSKs onto BMSCs in a dose-dependent manner (Supplementary Fig. 13).

Supplementary Figure 13. *In vivo* administration of GSK126 into WT mice mimics the phenotype of KO mice. WT mice were sacrificed on 16 h after intraperitoneal injection of GSK126 (50 mg/kg or 100 mg/kg) to monitor VCAM-1 expression in BMSCs and perform trans-stromal migration and adhesion assays. $n = 5$. **a**, **b** Relative ratio of VCAM-1 expression in BMSCs isolated from GSK126 injected mice was analyzed by immunoblot (**a**) and flow cytometry (**b**). EC, endothelial cells; OB, osteoblasts; MSC, mesenchymal stem cells. **c** Relative ratio of migrated LSK cells through BMSCs isolated from GSK126 injected mice. **d** Relative ratio of adhered LSK to BMSCs isolated from GSK126 injected mice. All data are presented as the mean \pm SEM. * $P < 0.05$; ** $P < 0.01$.

According to these changes, we revised the manuscript (line 242 in page 16 – line 248 in page 17, line 635 in page 42 – 642 in page 43).

8) Page 7, last 3 lines: Clarify how you know that this is decreased migration.

: We acknowledge your comment. According to your suggestion, we revised the manuscript (line 112 - 114 in page 8).

9) Page 8, line 6: Data in Figure 1g should be shown as absolute numbers of colonies in bone marrow (e.g. femur), and to add to this what were the absolute numbers of colony forming cells of each type (CFU-GM, BFU-E and CFU-GEMM) in PB and spleen?

: We acknowledge your comment. According to your suggestion, we showed the absolute numbers of colony forming cells of each type (CFU-GM, BFU-E and CFU-GEMM) in BM, PB and spleen in Supplementary Figure 1 of the revised manuscript.

10) Supplementary Figure 1: I assume that CFU-C refers to CFU-GM unless this is for all colonies so be consistent in nomenclature between Figures (e.g. Fig 1g), and why is this information not available for each progenitor cell type (CFU-GM, BFU-E and CFU-GEMM) also?

: We acknowledge your comment. CFU-C refers to all colonies (CFU-GM + BFU-E + CFU-GEMM). According to your suggestion, we showed the absolute numbers of colony forming cells of each type (CFU-GM, BFU-E and CFU-GEMM) in BM, PB and spleen in Supplementary Figure 1, Supplementary Figure 5 and Supplementary Figure 14 of the revised manuscript.

11) Figure 2d and the CFU-C/ml PB in Fig. 2e: These decreases, although significant are very modest with AMD3100. Since AMD3100 synergizes with G-CSF to mobilize stem and progenitor cells it would be important to assess this in context of mobilization with G-CSF and AMD3100 as has been reported by others.

: We acknowledge your comment. According to your suggestion, we performed G-CSF and AMD3100-induced mobilization assay. Results clearly indicated that Phc2 KO HSPC migration are significantly decreased compared with WT HSPC migration (Supplementary Figure 17 in the revised manuscript).

According to these changes, we added one reference and changed the revised version of manuscript (line 279 - 280 in page 19, lines 493 - 497 in page 33).

Supplementary Figure 17. Administration of anti-VCAM-1 Ab reverses the defect involving G-CSF and AMD3100-induced HSPC mobilization in KO mice. G-CSF and AMD3100-induced HSPC mobilization assay with anti-VCAM-1 Ab is described in “Methods” section. $n = 5$ per group. **a** WBC counts from experimental mice administered with G-CSF, AMD3100 and/or anti-VCAM-1 Ab. **b** Representative image of spleens (*left*) and absolute number of splenocytes (*right*) from experimental mice administered with G-CSF, AMD3100 and/or anti-VCAM-1 Ab. **c** CFU-C in the BM (femur, *left*), PB (*middle*), and spleen (SP, *right*) from experimental mice administered with G-CSF, AMD3100 and/or Ab. “-”, treatment with vehicle or isotype control Ab; “+”, treatment with G-CSF or anti-VCAM-1 Ab. All data are presented as the mean \pm SEM. * $P < 0.05$; ** $P < 0.01$.

Reference

60. Ramirez, P., et al. BIO5192, a small molecule inhibitor of VLA-4, mobilizes hematopoietic stem and progenitor cells. *Blood* **114**, 1340-1343 (2009).

Minor Comments

- 1) In Page 9 Line 137, “WT mice were significantly...” should be “KO mice were significantly...”. In addition, statistics are missing in Fig5f and g.

: Thank you very much for your kindness. We fixed sentence according to your suggestion (line 141 in page 10) and included statics in Figure 5f and Figure 5g of the revised manuscript.

2) Page 9, line 137: Shouldn't WT mice should be Phc2 KO mice?

: Thank you very much for your kindness. We fixed sentence according to your suggestion (line 141 in page 10).

Reviewer #3

Major Concerns

1) The rationale for studying the Phc2 in HSPC mobilization is not clear.

: We acknowledge your comment. According to your comment, we revised manuscript (line 106 - 108 in page 8).

2) In the introduction, the authors mentioned that “several BM niche proteins have been identified as key signaling components to control the maturation of HSPCs”. Some of the proteins in the niche critical for HSPC functions should be listed.

: We acknowledge your comment. According to your comment, we revised manuscript (line 76 - 77 in page 8).

3) The specific types of the CFUs should be clarified.

: We acknowledge your comment. CFU-C refers to all colonies (CFU-GM + BFU-E + CFU-GEMM). According to your suggestion, we showed the absolute numbers of colony forming cells of each type (CFU-GM, BFU-E and CFU-GEMM) in BM, PB and spleen in the revised manuscript (Supplementary Figure 1, Supplementary Figure 5 and Supplementary Figure 14).

4) What is the impact of the Phc2 deletion on the myeloid lineages in different tissue organs?

: Thank you for your constructive comment. According to reviewer's comment, we did additional experiments with peritoneal macrophages. As a result, *Vcam1* mRNA levels were significantly elevated in activated KO peritoneal macrophages compared to those of WT (Supplementary Fig. 11). According to this change, we revised manuscript (line 213 – 215 in page 15).

5) Figure 4, in an *in vivo* homing assay, the authors measured the number of CFSE+ cells homing to the BM. Since the CFSE+ cells may be a mixed population, more detailed characterization is needed to determine the specific lineage populations in the CFSE+ cell population.

: Thank you for your constructive comment. According to your suggestion, we did additional experiments using LSK cells for *in vivo* homing assay. Therefore, Figure 4 and Figure 7 are the results from *in vivo* homing assay using CFSE+ LSK cells in the revised manuscript.

According to these changes, we revised manuscript (line 181 – 185 in page 13, line 538 in page 36 – line 560 in page 37, line 904 – 908 in page 62, line 954 – 956 in page 65).

We also showed the absolute numbers of colony forming cells of each type (CFU-GM, BFU-E and CFU-GEMM) in BM, PB and spleen in the revised manuscript (Supplementary Figure 5).

6) In Figure 4c, the authors assessed the frequency of CFU-Cs in the BM, PB and spleen of the recipient mice. Again, specific types of the CFUs should be characterized.

: Thank you for your constructive comment. As mentioned above, we showed the absolute numbers of colony forming cells of each type (CFU-GM, BFU-E and CFU-GEMM) in BM, PB and spleen in the revised manuscript (Supplementary Figure 5).

7) In Figure 7 c,d, the authors quantified CFU-C and CFSE+ cells in an *in vivo* homing assay following the blockage of the interaction between VCAM-1 and VLA-4. The authors claimed that “blocking the interaction between VCAM-1 and VLA-4 could restore LSK cell mobilization from the KO BM into the PB and spleen”. The frequencies of specific HSC/HPC populations by flow cytometric analyses should be provided.

: Thank you for your constructive comment. As mentioned above, we did additional experiments using LSK cells for *in vivo* homing assay shown in Figure 7. We also showed the absolute numbers of colony forming cells of each type (CFU-GM, BFU-E and CFU-GEMM) in BM, PB and spleen in the revised manuscript (Supplementary Figure 14).

8) While the authors provided ChIP-qPCR data showing the Phc2 binding sites on the *Vcam1* gene, ChIP-seq may provide strong evidence for the role of Phc2 on H2AK119ub and H3K27 methylation-mediated gene regulation.

: Thank you for your constructive comment. We absolutely agreed with your opinion. Identification of universal Phc2 binding region is very helpful to identify target genes related to PcG protein function. According to your comment, we identified binding sites of Phc2 on the *Vcam1* locus (Figure 6a and Figure 6b). Also, we showed status of H3K4me3, H3k27me3, H2AK119ub in these regions comparing WT BMSCs to KO BMSCs (Figure 6c). Further, we provided evidence indicating that other components of PRC1 and PRC2 bind to the same region (Figure 6c). Therefore, these data provide strong evidence about the functional role of Phc2 as a regulator of *Vcam1* expression.

9) All the statistical data is based on the two-tailed Student's t test. However, since many of the study groups were four, statistical analyses with ANOVA would be more appropriate.

: Thank you for your constructive comment. According to your suggestion, we did statistical analyses with ANOVA in some figures in the revised manuscript (Figure 4, Figure 7, Supplementary Figure 5, and Supplementary Figure 14).

According to these changes, we revised manuscript (line 646 – 648 in page 43, line 912 – 913 in page 62, line 968 – 969 in page 66).

Minor concerns

1. Supplementary Fig. 3 in line 155 on page 10 should be Supplementary Fig. 4.

: We acknowledge your comment. According to your suggestion, we revised the manuscript (line 159 in page 11).

2. In Figure 5, the authors showed a higher level of Vcam1 in Phc2^{-/-} BMSCs due to a reduction of H3K27me3 and H2AK119ub levels. However, the authors stated that “Phc2 can directly repress Vcam1 expression through histone modifications”. The “directly” may not be the appropriate word.

: We acknowledge your comment. According to your suggestion, we revised the manuscript (line 249 in page 17).

3. Please do spelling and grammar check for the paper.

: We acknowledge your comment. Before submission, our manuscript was edited by professional English proofreading specialist through Nature Research Editing Service. Record number in Nature Research Editing Service is CC46-8FC5-8F35-1314-AA3P. Also, authors more carefully check spelling and grammar before submission.

REVIEWERS' COMMENTS:

Reviewer #1 (Remarks to the Author):

In this study, the authors show that Phc2, a component of the PRC1 complex, regulates HSPC trafficking by modulating VCAM1 expression on bone marrow stromal cells. This is a novel observation that provides important new insight into how epigenetic modifiers may regulate hematopoiesis in a non-cell extrinsic fashion. The data are in general convincing, and only minor concerns need to be addressed.

1. The authors conclude that AML3100-induced mobilization is impaired in KO mice. This is not supported by the data in Figure 2d,e. The decrease in CFU-C in the blood is seen both at baseline and after AMD3100 treatment. Fold changes are likely to be non-significant. The text should be modified to reflect this.
2. The addition of Suppl Fig 18 nicely addresses the question whether Phc2 works in a non-hematopoietic cell intrinsic fashion to regulated HSPC trafficking. However, the discussion of the results on page 19 is confusing and needs to be revised.
3. The bone sections showing co-staining of VCAM1 with the different stromal cell populations is still not convincing (Figure 5, Supplemental Figure 7). Either better quality images are needed or the data should be removed from the paper.

Reviewer #2 (Remarks to the Author):

Thank you for your extensive efforts to deal with my suggestions and to revise the paper.

Reviewer #3 (Remarks to the Author):

The authors responded adequately for most of the questions. However, there is one question remains unsolved. The previous question was "The rationale for studying the Phc2 in HSPC mobilization is not clear". However, the authors commented that "We initially observed macroscopic abnormalities in the thymus and peripheral lymphoid organs of Phc2^{-/-} (KO) mice¹⁴ compared to those of Phc2^{+/+} (wild-type, WT) mice". This still does not provide the rationale for studying the Phc2 in HSPC mobilization.

Response to Reviewers

We appreciate the reviewers for their helpful and constructive comments. We have carefully read through the reviewer's comments and changed the manuscript according to the reviewer's suggestions.

We hope that this revised manuscript strengthens the main conclusion (the extrinsic role of epigenetic factor on bone marrow niche as a novel important regulator of hematopoietic stem/progenitor cell mobilization) of the original manuscript.

For reviewer's convenience, we prepared two different versions of revised manuscript. One is an unmarked version and another is a marked version. In marked version, all changes in text are marked with yellow highlights.

Please find a point-by-point responses to each of the reviewers' comments/questions along with additional data to support our findings. All corrections are detailed below.

Specific remarks

Reviewer #1 (Remarks to the Author):

In this study, the authors show that Phc2, a component of the PRC1 complex, regulates HSPC trafficking by modulating VCAM1 expression on bone marrow stromal cells. This is a novel observation that provides important new insight into how epigenetic modifiers may regulate hematopoiesis in a non-cell extrinsic fashion. The data are in general convincing, and only minor concerns need to be addressed.

: We truly appreciate your generous and valuable comments. According to your suggestion, we revised the manuscript. Below, you will find the point-by-point responses in regards to individual queries. Thank you very much.

1. The authors conclude that AML3100-induced mobilization is impaired in KO mice. This is not supported by the data in Figure 2d, e. The decrease in CFU-C in the blood is seen both at

baseline and after AMD3100 treatment. Fold changes are likely to be non-significant. The text should be modified to reflect this.

: Thank you for your constructive comment. When we used an AMD3100 as a mobilizing agent of LSK, we still observed that the migration of KO LSK into the periphery is statistically decreased compared with that of WT LSK. However, the defective migration of KO LSK after treatment of AMD3100 is relatively modest when we compared with results from G-CSF treatment. According to your suggestion, we revised the manuscript (line 146 in page 10 – line 150 in page 11).

2. The addition of Suppl Fig 18 nicely addresses the question whether Phc2 works in a non-hematopoietic cell intrinsic fashion to regulated HSPC trafficking. However, the discussion of the results on page 19 is confusing and needs to be revised.

: Thank you very much for your kindness to point out our mistake. According to your suggestion, we fixed sentence regarding the description of Suppl Fig 18 in the revised manuscript (line 288 – 293 in page 20).

3. The bone sections showing co-staining of VCAM1 with the different stromal cell populations is still not convincing (Figure 5, Supplemental Figure 7). Either better quality images are needed or the data should be removed from the paper.

: Thank you for your constructive comment. According to your comment, we tried to make better quality images several times. Unfortunately, we failed to make better quality images. Therefore, we removed figures displaying the bone sections showing co-staining of VCAM1 with the different stromal cell populations (Figure 5c and Supplemental Figure 7) in the revised manuscript.

Reviewer #2 (Remarks to the Author):

General Comments

Thank you for your extensive efforts to deal with my suggestions and to revise the paper.

: We really appreciate your kind comments. Your previous comments have been very helpful in improving the quality of the manuscript.

Reviewer #3 (Remarks to the Author):

The authors responded adequately for most of the questions. However, there is one question remains unsolved. The previous question was “The rationale for studying the Phc2 in HSPC mobilization is not clear”. However, the authors commented that “We initially observed macroscopic abnormalities in the thymus and peripheral lymphoid organs of Phc2^{-/-} (KO) mice¹⁴ compared to those of Phc2^{+/+} (wild-type, WT) mice”. This still does not provide the rationale for studying the Phc2 in HSPC mobilization.

: We truly appreciate your generous and helpful comments. According to your comment, we revised the text as following sentence: “As an initial step to elucidate the functional role of Phc2 during hematopoiesis, we characterized immune phenotypes of *Phc2*^{-/-} (KO) mice¹⁴ compared to those of *Phc2*^{+/+} (wild-type, WT) mice.” (line 105 – 107 in page 8 of the revised manuscript).